# Rapid detection of microbiota cell type diversity using machine-learned classification of flow cytometry data

Birge D. Özel Duygan[1✉], Noushin Hadadi [1,3], Ambrin Farizah Babu[1], Markus Seyfried[2] &
Jan R. van der Meer [1✉]

The study of complex microbial communities typically entails high-throughput sequencing and downstream bioinformatics analyses. Here we expand and accelerate microbiota analysis by enabling cell type diversity quantification from multidimensional flow cytometry data using a supervised machine learning algorithm of standard cell type recognition (CellCognize). As a proof-of-concept, we trained neural networks with 32 microbial cell and bead standards. The resulting classifiers were extensively validated in silico on known microbiota, showing on average 80% prediction accuracy. Furthermore, the classifiers could detect shifts in microbial communities of unknown composition upon chemical amendment, comparable to results from 16S-rRNA-amplicon analysis. CellCognize was also able to quantify population growth and estimate total community biomass productivity, providing estimates similar to those from [14]C-substrate incorporation. CellCognize complements current sequencing-based methods by enabling rapid routine cell diversity analysis. The pipeline is suitable to optimize cell recognition for recurring microbiota types, such as in human health or engineered systems.

[1] Department of Fundamental Microbiology, University of Lausanne, 1015 Lausanne, Switzerland. [2] Biotechnology Department, Firmenich SA, Geneva, Switzerland. [3]Present address: Department of Cell Physiology and Metabolism, Faculty of Medicine, University of Geneva, CH-1211 Geneva, Switzerland. ✉email: birgeozel@gmail.com; Janroelof.vandermeer@unil.ch

With the increasing realization of the crucial roles played by microbial communities for human[1], animal[2] and plant health[3], and for biogeochemical processes in the environment[4], high-throughput methods of microbiota analysis are gaining ever greater importance. Current methods are largely "omics"-based and emphasize taxonomic[5,6] or functional gene diversity[7]. Despite their great reliability and sensitivity, omics-based approaches are still relatively slow and laborious, which is a disadvantage in fields requiring rapid expert decisions, such as for clinical interventions. In addition, they frequently or inherently underestimate absolute population densities[8,9] and neglect variation in microbial cell physiologies within the microbiota[10]. Stable isotope-labeled substrate incorporation, coupled with metagenomic tools[11,12], can inform about specific substrate use by species within a community, but stable isotopes are too expensive to be deployed for routine microbiota analyses. Growth of individual species within microbiota may be further inferred from individual cell mass measurements[13] or indirectly, from binned metagenomic sequence read coverage differences[14], but neither is simple for rapid routine microbiota analysis.

There is thus a clear need for methods to complement and expand current omics-dominated microbiota analyses, which we propose might be accomplished by high-throughput single-cell analyses based on flow cytometry (FCM). The main advantages of FCM are its simplicity and sensitivity, and by providing absolute counts of suspended cells, it enables real-time sample analysis and interpretation[15]. Cells are detected in FCM on the basis of optical properties (light scatter from cell shape and structures)[16], and can further be stained with a plethora of fluorescent dyes that target specific biomolecules (e.g., nucleic acids)[17] or physiological activity (e.g., membrane permeability to distinguish viable from compromised cells)[18–20]. However, despite the ease with which a wide variety of FCM parameters can be recorded on large numbers of individual cells, there is no straightforward relation between the multidimensional FCM data and the identity of bacterial strains or cell properties, particularly within diverse microbiota. Some success has been achieved in inferring microbiota compositional changes from FCM data using unsupervised clustering or cytometric fingerprinting[21–23], but without species recognition[24,25]. However, species recognition from FCM data has proved possible for freshwater and marine unicellular eukaryotes, likely because of their larger size[26], and recent multiparametric statistical studies suggest that, in principle, even closely related bacterial strains can be differentiated from FCM data[27]. Machine learning provides an efficient and versatile approach to extract a classification from complex data, but has so far not been applied to microbiota classification, except for synthetic mixtures of bacterial species[16,28], or as a support for diversity analysis by high-throughput amplicon sequencing[29,30].

Here we present a new pipeline to facilitate microbiota diversity analysis by providing rapid absolute quantification of cells and recognition of cell type diversity and physiology based on flow cytometry data. We named this pipeline CellCognize. In contrast to recent FCM fingerprinting and single-cell classification approaches[16,27,28,30], CellCognize analyzes multiparametric FCM data by comparison to a set of predefined microbial and bead size standards, which the program is trained to recognize using a supervised artificial neural network (ANN) (Fig. 1). We provide proof of concept of the CellCognize pipeline by first quantifying species composition in synthetic mixtures of known bacterial strains and in mixtures of known bacterial strains within a diverse background of unknown microorganisms. We then test to what extent CellCognize can analyze microbial cell diversity of unknown communities and changes thereof, imposed by addition of low concentrations of phenol or 1-octanol, which we compare with data from 16S rRNA gene amplicon diversity analysis and

further interpret based on the probability of class assignment. Finally, we estimate community growth from CellCognize data of substrate-amended unknown microbiota, and compare this to biomass productivity from [14]C-labeled substrate incorporation. Our results demonstrate the ability to recognize and quantify known microbial cell types, their physiology and growth, amidst a known or unknown community background, and even infer community diversity changes in unknown microbial communities. This suggests that with appropriate standards and optimization, CellCognize can reliably analyze recurring microbiota cell types in environmental, animal or clinical settings, thereby considerably accelerating and simplifying microbiota studies.

## Results

**Development of an artificial neural network categorizing microbial cell types from multiparametric flow cytometry data.** We developed a pipeline (CellCognize) using a supervised artificial neural network (ANN), which classifies cell types in microbial community samples based on FCM multiparametric signature similarities with a predefined set of standards (Fig. 1). FCM signatures of the standards are first captured individually (Fig. 1a,b), then combined in silico to build the training, validation and testing sets, which the network learns to differentiate in a feed-forward back-propagation algorithm (Fig. 1c, "Methods", Supplementary Methods). The output of the trained, validated and tested ANN model is a set of learned linear equations (*classifier*, for this and other used terminology, see Supplementary Notes). The classifier can then be used to assign each cell within untrained samples (Fig. 1d) on the basis of its FCM signature into its most similar output class (Fig. 1e), and calculate relative abundances or biomass of that standard in the community (Fig. 1f). Predicted classifications come with a corresponding probability score, which we envisioned may be interpreted as a measure of similarity to the standard. This might be useful when analyzing the cell diversity of samples from unknown microbiota in which the standard strains used to build the classifiers are absent (Fig. 1d).

**Differentiating and categorizing microbiota of known composition.** To test the conceptual idea, we first assembled and classified a synthetic community consisting of the three bacterial species *Escherichia coli*, *Pseudomonas veronii*, and *Acinetobacter johnsonii*. FCM signatures of individual cultures stained with SYBR Green I were captured in seven channels, filtered and gated to five classes (both *E. coli* and *P. veronii* yielded two visible subpopulations in FCM, see "Methods", Supplementary Fig. 1, Supplementary Methods, Section 3.1). Next, in silico merged FCM data sets were used to train the ANN. The network differentiated the five classes with a mean precision and recall of 81% (Supplementary Fig. 2). The ANN-5 classifier assigned 76–88% of cells in experimentally regrown pure cultures to the correct species (i.e., correct predicted classification, see Supplementary Notes for definition of terms). In addition, the correct predicted classification of cells in defined three-species mixtures was between 96% and 132% (Fig. 2a, Supplementary Methods, Section 3.2–3.3).

To test the approach for more complex communities of known composition, we expanded to a set of 32 standards consisting of eight polystyrene bead standards of different diameter, one yeast culture, and fourteen bacterial strains (Supplementary Table 1), of which six had two distinguishable subpopulations in FCM data and one had three (Table 1, Supplementary Fig. 1). The choice of standards was arbitrary but initially motivated by (i) a priori cell type and size (e.g., rod, coccus) or bead size differences (Supplementary Fig. 3), (ii) the potential presence of similar

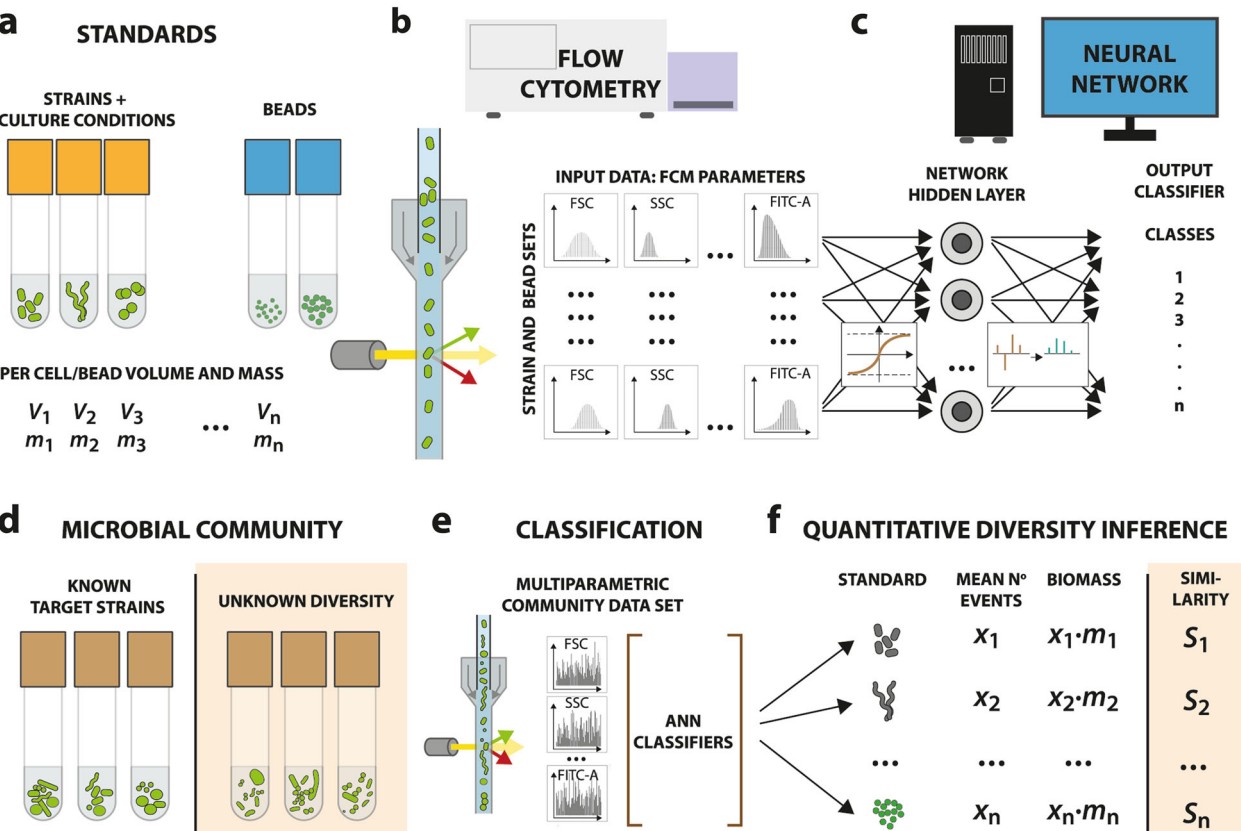

**Fig. 1 CellCognize: a flow cytometry (FCM)— supervised artificial neural network (ANN) pipeline for classification of microbial cell diversity and physiology.** Representative stained cell and bead standards with known volume and mass (**a**) are analyzed by FCM to capture multidimensional optical and shape characteristics (**b**). Note that FITC here represents the channel to capture the SYBR Green I fluorescence of cell staining. Multiparametric data of each of the strain and bead standards, separated where they consist of recognizable subpopulations, are used as input for training, testing and validating the ANN, producing the classifiers (**c**). FCM data from stained target untrained known or unknown microbial communities (**d**) are assigned to the strain and bead output classes using the ANN classifiers (**e**). The diversity attribution can subsequently be used to estimate individual population densities and their biomass, and, in the case of unknown communities, to calculate similarities to the used standards (**f**).

strains in our target freshwater microbial community, and (iii) the inclusion of multiple representatives from the same genus (e.g., *Pseudomonas*, *Sphingomonas*) or species (e.g., *E. coli* MG1655 and DH5α-λpir). FCM signatures of the 32 standards were distinct in principal component analysis (PCA, based on $n$ = 20,000 cells per standard), with two PCA components explaining >90% of the covariation (Fig. 2b, Supplementary Methods, Section 3.4).

ANNs were trained with in silico merged multiparametric FCM data sets consisting of each of the 32 standards (randomly subsampled to the same size, $n$ = 10,000 before merging; Supplementary Methods, Section 2.2–2.3). This process was repeated five times independently, resulting in five slightly different ANN-32 classifiers. When used to classify additional in silico merged FCM datasets of the standards (not those used for training), these classifiers achieved a mean accuracy of 79.2% (range 27.3–99.8% across the 32 standards, Fig. 2c, Supplementary Fig. 2, Table 1), and with 80–99% true positive identification at <20% false positives (Supplementary Fig. 2, Supplementary Methods, Section 2.1–2.3). The precision and recall rates varied among the standards, with beads on average classified more accurately than strain standards, in accordance with their greater separation in the PCA (Fig. 2c, Table 1). Beads and strains were rarely misclassified with each other, but some strain standards were reciprocally confused, whereas others were very consistently differentiable based on their FCM signatures (Table 1, Supplementary Fig. 2, Supplementary Dataset 1). Although this was not

tested extensively, confusion was not dependent on standards being taxonomically closely related. For example, several *Pseudomonas* strains were well distinguished (Supplementary Fig. 2). Neither were intuitive cell shape differences an obvious differentiation criterion. For example, although the larger *Bacillus subtilis* rods (BST1) were well differentiated from all other rod-shaped bacteria (mostly *Pseudomonas* standards, Table 1), the curved cells of *Caulobacter crescentus* (Supplementary Fig. 2, CCR1) were confused to some extent with the small rod-shaped *Pseudomonas putida* (PPT) and with the irregularly shaped cells of *Arthrobacter chlorophenolicus* (ACH, Supplementary Fig. 2). These tests indicated that CellCognize is able to differentiate a set of 32 standards from each other based on their multiparametric FCM signatures, albeit with precision and recall that varied among the standards. Some of the weaker differentiation might be due to cell heterogeneity within single standards, or unresolved similarities in cell morphology and optical characteristics between standards based on the employed FCM parameters and staining.

**Differentiation of cell physiology among *E. coli* strains.** To determine the potential of CellCognize to differentiate among closely related strains and different growth phases, we included among our standards two *E. coli* strains (MG1655 and DH5α-λpir). The two *E. coli* strains were grown to stationary phase on LB medium (MG_STAT_LB or DH5_STAT_LB), while MG1655 was further sampled in exponential (MG_EXP) and stationary phase on M9-CAA medium (MG_STAT_MM). Strikingly, the

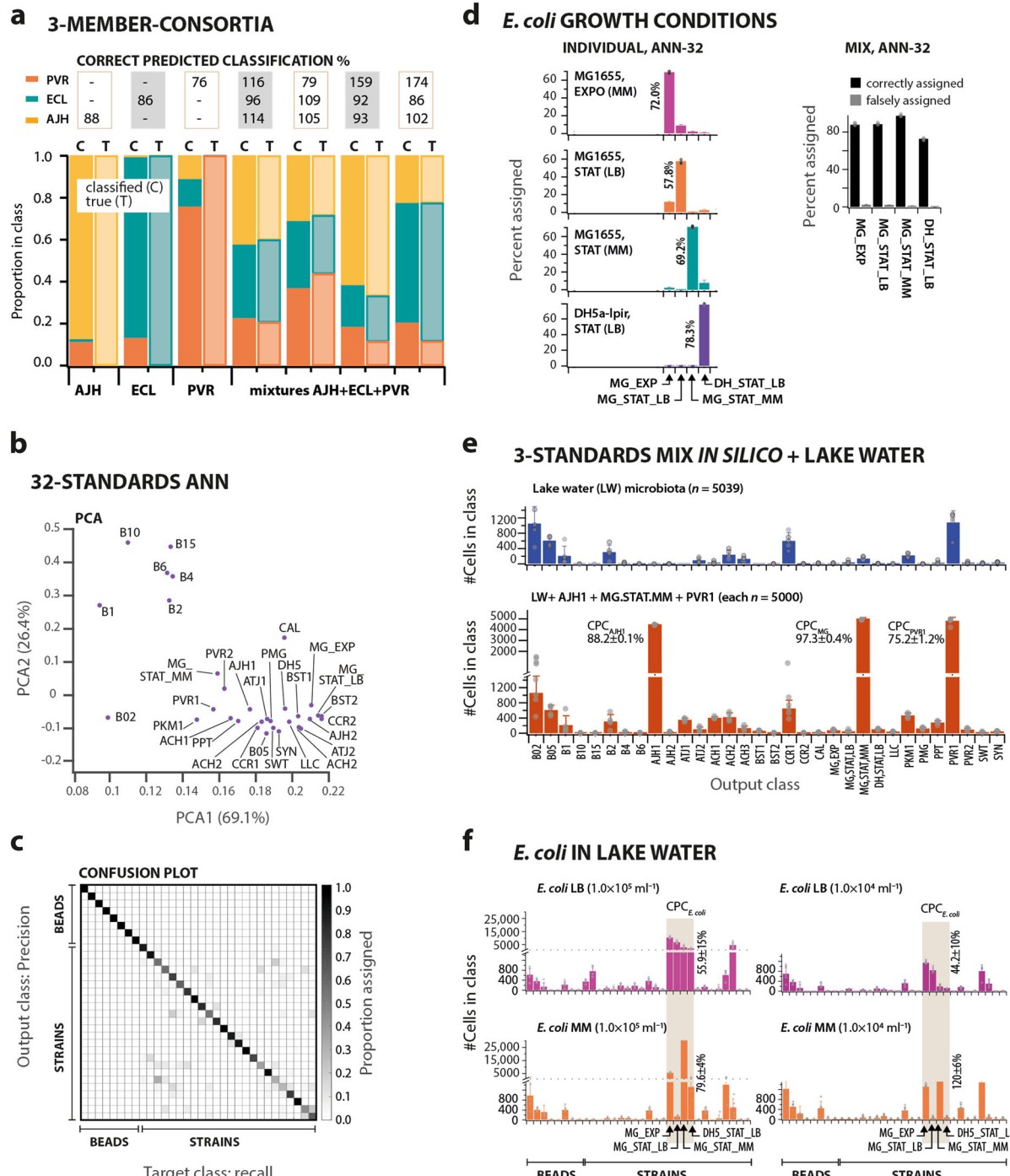

32-standard ANN classifiers correctly predicted classification of 58–78% of cells in the experimental datasets of each of the four *E. coli* cultures individually, and 70–90% of an in silico mixed FCM dataset (Fig. 2d, Supplementary Methods, Section 3.5–3.6). Among these four standards, it was possible to clearly differentiate cells according to growth phase (strain MG1655 at exponential phase on M9-CAA medium, MG_EXP vs. stationary phase on M9-CAA medium, MG_STAT_MM) and culture medium (strain MG1655 at stationary phase on LB medium, MG_STAT_LB vs. stationary phase on M9-CAA medium, MG_STAT_MM), and to distinguish between closely related strains even when sharing the same growth phase and culture medium (strain MG1655 at stationary phase on LB medium, MG_STAT_LB vs. strain DH5α-λpir at stationary phase on LB

medium, DH5_STAT_LB, Fig. 2c). There is thus a great potential to determine cell physiological status and differentiate among closely related strains on the basis of FCM signatures using machine learning algorithms for recognition.

**Recognition of known standards within a diverse unknown aqueous microbial community.** To test the performance of CellCognize to recognize known strains within a complex microbiota, we assessed its ability to correctly predict classification of the 32 standards within a background of unknown microbes. We first assessed performance in silico by merging a randomly subsampled FCM dataset with 5000 events (not those used for ANN training) from each of the individual strain and bead standards separately with the same number of unknown

**Fig. 2 CellCognize performance and analysis of microbiota with known members. a** Classification of a three-membered bacterial community composed of *Acinetobacter johnsonii* (AJH), *Escherichia coli* MG1655 (ECL), and *Pseudomonas veronii* (PVR), using a five-class ANN classifier. Bars show the means of CellCognize-inferred strain abundance for in vivo grown pure cultures and mixtures compared to their true abundance, with correct predicted classification per strain indicated above. **b** Principal component analysis of multiparametric variation among the 24 defined cell and 8 bead standards (7 FCM parameters; 20,000 events for each), and the confusion matrix (**c**) for the 32-standard ANN classifiers showing the mean precision (rows) versus recall (columns), represented as gray-level, according to the scale bar on the right. **d** Correct prediction classification of *E. coli* MG1655 or DH5α-λpir cultures grown to exponential (EXPO) or stationary phase (STAT) in M9-CAA (MM) medium or in Luria broth (LB), individually (left, $n = 20,000$ cells) or as an in silico mixture (right, $n = 5000$ cells each, randomly subsampled). Bar plots show the mean class attribution ± one SD and together with the correct predicted classification of *E. coli*, from five independent ANN-32 classifiers. **e** Predicted classification (absolute cell counts ± one SD) from the five 32-standard ANN classifiers for cells from a Lake Geneva microbial community (blue bars, $n = 5039$) or for the same community in silico mixed with $n = 5000$ cells each of the standards AJH1, MG_STAT_MM and PVR1 (dark orange bars). Correct predicted classifications (CPC) were calculated as the mean percentage of each standard attributed to its own class. **f** Predicted classification (mean of absolute cell counts ± one SD, five 32-standard ANN classifiers) of triplicate FCM data of in vivo filtered (0.2–40 μm) Lake Geneva microbiota mixed with $1.0 \times 10^4$ or $1.0 \times 10^5$ cells ml$^{-1}$ of *E. coli* strain MG1655 grown on LB or M9-CAA medium (MM) to stationary phase. Correct predicted classifications (CPC) were calculated as the mean number (±one SD) of cells assigned to the four *E. coli* classes as a percentage of the expected added number.

## Table 1 Figures of merit for the standard cell classification.

| Standard abbreviation | Full name | remark | Percentage (Mean ± st dev)[a] | | |
|---|---|---|---|---|---|
| | | | Recall[b] | Precision[b] | Correct predicted classification in LW[c] |
| B02 | Beads 0.2 μm | | 98.3 ± 0.1 | 99.0 ± 0.1 | 98.5 ± 0.5 |
| B05 | Beads 0.5 μm | | 99.4 ± 0.1 | 98.7 ± 0.1 | 99.3 ± 0.5 |
| B1 | Beads 1 μm | | 99.8 ± 0.1 | 99.6 ± 0.1 | 99.8 ± 0.3 |
| B2 | Beads 2 μm | | 99.4 ± 0.2 | 99.6 ± 0.2 | 99.3 ± 0.3 |
| B4 | Beads 4 μm | | 96.3 ± 0.3 | 97.8 ± 0.5 | 96.3 ± 0.5 |
| B6 | Beads 6 μm | | 98.1 ± 0.4 | 96.5 ± 0.3 | 97.6 ± 0.6 |
| B10 | Beads 10 μm | | 98.8 ± 0.3 | 99.8 ± 0.1 | 99.1 ± 0.3 |
| B15 | Beads 15 μm | | 99.8 ± 0.0 | 98.9 ± 0.3 | 99.7 ± 0.2 |
| AJH1 | *Acinetobacter johnsonii* | subpop 1 | 88.7 ± 0.6 | 79.2 ± 0.5 | 88.5 ± 1.7 |
| AJH2 | | subpop 2 | 90.9 ± 1.0 | 77.9 ± 0.8 | 90.7 ± 2.0 |
| ATJ1 | *Acinetobacter tjernbergiae* | subpop 1 | 56.2 ± 2.1 | 47.0 ± 1.0 | 57.9 ± 0.9 |
| ATJ2 | | subpop 2 | 58.8 ± 3.8 | 59.5 ± 2.7 | 59.1 ± 0.5 |
| ACH1 | *Arthrobacter chlorophenolicus* | subpop 1 | 72.7 ± 1.1 | 56.8 ± 0.8 | 74.8 ± 0.7 |
| ACH2 | | subpop 2 | 63.4 ± 1.3 | 66.1 ± 1.9 | 63.5 ± 2.4 |
| ACH3 | | subpop 3 | 78.2 ± 0.9 | 70.6 ± 2.4 | 74.1 ± 3.2 |
| BST1 | *Bacillus subtilis* | subpop 1 | 97.8 ± 0.3 | 95.7 ± 4.3 | 92.9 ± 0.2 |
| BST2 | | subpop 2 | 80.8 ± 0.9 | 76.6 ± 1.1 | 81.2 ± 1.4 |
| CCR1 | *Caulobacter crescentus* | subpop 1 | 54.0 ± 2.0 | 62.0 ± 1.9 | 53.2 ± 0.1 |
| CCR2 | | subpop 2 | 79.5 ± 1.9 | 83.1 ± 1.2 | 78.3 ± 4.7 |
| CAL | *Cryptococcus albidus* | | 99.9 ± 0.0 | 99.8 ± 0.1 | 99.8 ± 2.0 |
| MG_EXP3 | *Escherichia coli* MG1655 | exponential phase | 88.2 ± 0.6 | 87.5 ± 1.1 | 87.8 ± 0.5 |
| MG_STAT_LB | | stationary phase LB | 89.3 ± 1.0 | 90.0 ± 0.7 | 88.7 ± 1.3 |
| MG_STAT_MM | | stat phase M9-CAA | 97.4 ± 0.8 | 96.7 ± 0.8 | 97.7 ± 1.8 |
| DH_STAT_LB | *Escherichia coli* DH5α-λpir | | 73.0 ± 0.9 | 83.5 ± 1.1 | 72.6 ± 0.6 |
| LLC | *Lactococcus lactis* | | 34.0 ± 3.3 | 49.9 ± 3.0 | 34.8 ± 1.8 |
| PKM1 | *Pseudomonas knackmussii* | | 94.0 ± 0.8 | 87.7 ± 0.7 | 93.6 ± 1.1 |
| PMG | *Pseudomonas migulae* | | 32.9 ± 2.8 | 39.5 ± 3.9 | 32.6 ± 3.1 |
| PPT | *Pseudomonas putida* | | 27.3 ± 4.0 | 38.2 ± 3.4 | 27.5 ± 3.5 |
| PVR1 | *Pseudomonas veronii* | subpop 1 | 73.9 ± 0.7 | 77.2 ± 1.9 | 74.3 ± 4.9 |
| PVR2 | | subpop 2 | 96.7 ± 0.7 | 96.7 ± 0.7 | 96.8 ± 0.7 |
| SWT | *Sphingomonas wittichii* | | 44.1 ± 1.5 | 52.7 ± 3.4 | 44.4 ± 1.5 |
| SYN | *Sphingomonas yanoikuyae* | | 66.6 ± 1.2 | 55.6 ± 1.0 | 65.0 ± 2.2 |

[a]Calculated from the five independently built ANN classifiers.
[b]See Supplementary Notes for terminology.
[c]Mean percentage ± one SD of each individual standard ($n = 5000$ subsampled cells) in silico mixed to a background of a lake water microbial community ($n = 5039$), attributed to its own class.

cells from a freshwater microbial community (Supplementary Fig. 4, Supplementary Methods, Section 3.9–3.10). These merged datasets were classified independently using the five ANN-32 classifiers and the percentage of correctly classified cells was calculated. The correct predicted classification for each of the in silico merged standards in the presence of the unknown freshwater microbial community (Supplementary Fig. 4) was similar to that of the standards alone, showing that the presence of cells from unknown species does not interfere with recognition of the standards (Supplementary Fig. 2, Table 1). We then merged in silico FCM datasets of three standards (each subsampled to $n = 5000$ cells) simultaneously with the lake water microbiota background ($n = 5039$ cells) and classified the mixture using the five ANN-32 classifiers. The classifications of three standard strains

were correctly predicted at between 75.2% and 97.3%, demonstrating good recognition and differentiation (Fig. 2e, Supplementary Methods, Section 3.8–3.9).

We further experimentally tested the performance of CellCognize to distinguish known strains within a background of unknown freshwater microbes. For this, we chose *E. coli*, whose classification was correctly predicted with 73–97% within the in silico merged data (Table 1). *E. coli* MG1655 was grown to stationary phase on either M9-CAA or LB medium and mixed with the freshwater microbial community at $1.0 \times 10^4$ or $1.0 \times 10^5$ cells ml$^{-1}$, which was analyzed by FCM after 1–2 h (Fig. 2f, Supplementary Methods, Section 3.10). The lake water community itself had few cells attributed to the *E. coli* classes (Fig. 2e, blue bars), and the *E. coli* classes increased as expected upon experimentally adding *E. coli* MG1655 cells (Fig. 2f, grey shaded zones). Added *E. coli* cells were to a large extent classified to the category of their pre-culture signature (e.g., cells grown on M9-CAA classified to MG_EXP and MG_STAT_MM, Fig. 2f, orange bars), although a small proportion may have shown a physiological reaction to the change from the preculture medium to artificial lake water (e.g., added cells in stationary phase being classified as EXP, Fig. 2f). Based upon the expected numbers of *E. coli* cells, their correct predicted classification was 79.6–120% for M9-CAA, but 44.2–55.9% for LB-grown cells (Fig. 2f, Supplementary Methods, Section 3.10). These results indicated that CellCognize can identify and quantify specific focal strains and their physiological signature within complex microbiota mixtures.

**Analysis of unknown microbiota.** We envisioned that CellCognize could also potentially be applied to differentiate cell type diversity of unknown microbial communities in which none of the learned standards are necessarily present. This application may be useful as a rapid estimate of diversity to compare habitats, or changes in a microbiota between individuals or upon treatment. As diversity measure one could rely on assigned class abundances with respect to the set of predefined standards, while realizing that this is different from directly measuring microbial taxa diversity. To test the relevance of such an approach, we analyzed community changes after exposure to selective chemical compounds, which we quantified by CellCognize classification and 16S rRNA-gene amplicon sequencing diversity analysis. We further measured specific biomass production using $^{14}$C-labeled substrate and compared this to estimates based on the summed biomass from predicted classifications, as conceptually outlined in Fig. 1e, f.

In order to induce changes in the freshwater microbial community composition, we cultured the lake water samples in solutions amended with low concentrations of 1-octanol or phenol (0.1, 1, and 10 mg C l$^{-1}$). As expected, exposure to phenol or 1-octanol caused a rapid and profound change in the total community cell count, to an extent dependent on the added substrate and its concentration (Fig. 3a, abs. counts, Supplementary Fig. 5). Classification using CellCognize revealed an obvious shift in the community composition after only one day following amendment with 10 mg C l$^{-1}$ phenol (Fig. 3b rel. counts, Supplementary Methods, Section 4.1), culminating in growth and domination of cell types similar to the *Acinetobacter* standards (AJH1, AJH2, ATJ1, ATJ2) as well as *Pseudomonas migulae* (PMG) after two and three days, contributing 70% of the cells in the community (Fig. 3b, Supplementary Fig. 5). This was noticeably different from the detected change over time in the un-amended controls, whereas similar enrichments to the *Acinetobacter* classes were seen after amendment with 0.1 and 1 mg C l$^{-1}$ phenol at day 3 (Fig. 3b, rel. counts). Independent replicates of

phenol amendment to the Lake Geneva water microbial community in different months showed similar cell types becoming enriched (Supplementary Fig. 6). Similarly, amendment with 10 mg C l$^{-1}$ 1-octanol also caused a rapid increase in total community cell count in comparison to the un-amended controls (Fig. 3a, abs. counts, Supplementary Fig. 5). In this case, however, enriched cell types were more diverse and comprised various classes, none of which exceeded 15% of the total community (Fig. 3a, 1-octanol).

To qualify the performance of CellCognize in tracking community shifts, we compared in a separate experiment both CellCognize and molecular diversity analysis using 16S rRNA gene amplicon sequencing for 10 mg C l$^{-1}$ phenol and 1-octanol amendments (Fig. 3c). Both methods showed obvious strong enrichments in the substrate-amended lake water samples after three days, in a consistent manner across biological replicates (Fig. 3c, bracketed zones in stackplots). 16S rRNA gene amplicon analysis showed a strong decrease in richness in day 3 samples, which was not seen in CellCognize. Shannon diversity was moderately correlated between both methods ($r^2 = 0.5767$), but both methods grouped replicates, treatments and time effect equally well (Fig. 3c, MDS plots, ADONIS, $p < 0.001$). Bray–Curtis distances of the data sets from CellCognize or 16S rRNA gene amplicon analysis were similar (procrustes goodness-of-fit = 0.2144, Pearson-ranked correlation coefficient = 0.8981, $p = 0.0000$). This showed that although the underlying diversity measures differ between CellCognize and 16S rRNA gene amplicon sequencing, broad changes in communities can be captured equally well.

To further assess the value of CellCognize quantification of cell type diversity in unknown communities, we calculated biomass yields of the lake water microbial community upon phenol or 1-octanol amendment (Fig. 3a, abs. counts, Supplementary Fig. 5), using estimated respective standard per particle biomasses (Supplementary Table 2). These estimates were compared to independently measured biomass yields from triplicate assays for $^{14}$C-labeled substrate incorporation (Supplementary Fig. 7). Biomass yields were largely comparable (Table 2), although CellCognize estimates were in general lower, except at the lowest substrate concentration. This showed that the class enrichments deduced by CellCognize translate into reasonable biomass predictions even in unknown communities, which is support for the conclusion that enriched bacterial cell types are similar to the attributed standard classes.

**Similarity assessment of unknown microbiota and predefined CellCognize classes.** Given the large observed taxonomic diversity in the freshwater communities (Fig. 3c, 16S rRNA amplicon), a rightful question is what the assignment of unknown microbiota into the predefined standard classes in CellCognize actually means. In order to address this question we analyzed in greater depth the probabilities of class assignments for the standards themselves and for the unknown classified microbiota. Furthermore, we purified one isolate from the lake water substrate enrichments and compared its class assignments with the ANN-32 classifier and with a newly trained classifier that included that isolate.

In all CellCognize results so far (e.g, Figs. 2e and 3a), we were adopting a simple assignment criterion, assigning each cell to the class that yielded the highest probability of cell assignment (see, e.g., Supplementary Dataset 2, Supplementary Methods, Section 5.1). Although this procedure assigns cells to their most likely class, their probability score could still be lower than the mean score for cells from the standard itself. To illustrate this, we calculated the mean probabilities per assigned class for the

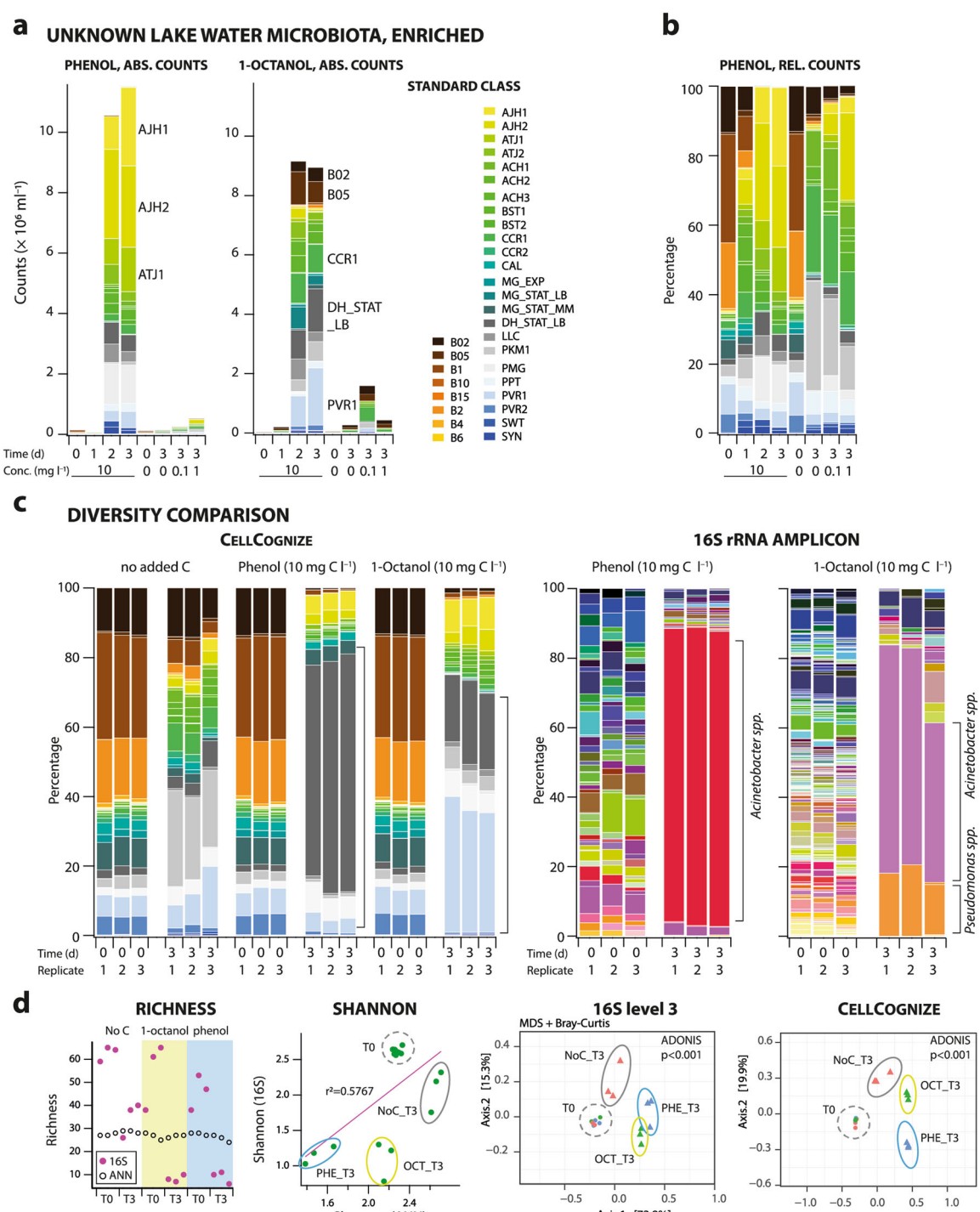

**Fig. 3 Diversity analysis of an unknown microbial community using CellCognize. a** Inferred mean class cell densities from the five 32-standard classifiers (absolute counts, ABS.) of a size-filtered (0.2–40 μm), resuspended Lake Geneva water microbial community over the course of three days amended with 0.1, 1 or 10 mg C l$^{-1}$ phenol or 1-octanol, compared to a zero added carbon control. Bars show individual biological replicates, with data merged from two technical replicates. **b** Proportional cell counts (REL.) for the phenol-amended communities shown in a. **c** Comparison of community diversity inferred using CellCognize and taxonomic diversity estimated from 16 S rRNA gene amplicon data (shown as proportions of 20,000 normalized cleaned sequence reads, given without color scale) for communities amended with 10 mg C l$^{-1}$ phenol or 1-octanol. **d** Diversity measures of communities shown in **c**: richness (16S: class level; CellCognize: assigned classes >0.05%) initially (T0) and after three days incubation (T3), Shannon index and Multidimensional scaling plot (MDS), based on calculated Bray–Curtis similarities. Symbols represent individual replicate diversities, circumscribed by ellipses to indicate similar treatments.

**Table 2 Comparative biomass yield estimates from [14]C-labeled substrate incorporation and from CellCognize for the Lake Geneva microbial community after 3 days incubation with phenol or 1-octanol as sole carbon sources at varying concentrations.**

| Substrate | Concentration (mg C l$^{-1}$) | [14]C biomass yield (g/g)[a] | CellCognize biomass yield (g/g)[a,b] | t-test |
|---|---|---|---|---|
| Phenol[c] | 0.1 | 0.135 ± 0.027 | 0.350 ± 0.15 | $p = 0.0709$ |
| | 1.0 | 0.151 ± 0.052 | 0.057 ± 0.010 | $p = 0.0370$ |
| | 10 | 0.166 ± 0.042 | 0.118 ± 0.017 | $p = 0.1393$ |
| 1-octanol[d] | 0.1 | 0.469 ± 0.168 | 0.367 ± 0.117 | $p = 0.4341$ |
| | 1.0 | 0.396 ± 0.024 | 0.148 ± 0.038 | $p = 0.0007$ |
| | 10 | 0.233 ± 0.028 | 0.100 ± 0.024 | $p = 0.0033$ |

[a]Mean ± one SD
[b]Calculated using mean per cell biomass for each standard as of Supplementary Table 2.
[c]Three independent experiments using lake water sampled on different occasions, with three biological replicates each.
[d]Single experiment with biological triplicates.

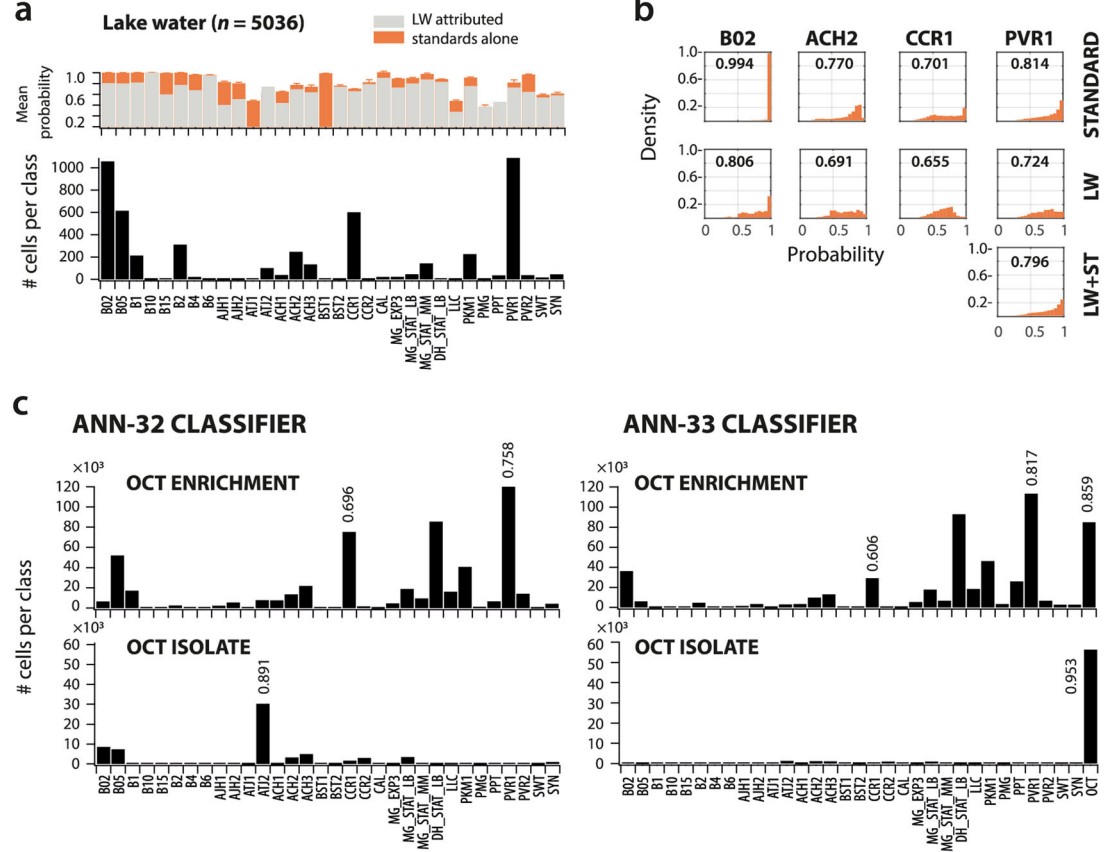

**Fig. 4 Similarity measures of cells attributed to CellCognize classes. a** Class attribution (absolute cell counts) from a single 32-standard ANN classifier for in vivo filtered (0.2–40 µm) $n = 5036$ cells from a Lake Geneva microbial community (black bars), with their corresponding mean probability of assignment (gray bars, LW attributed). In background (orange bars), mean probabilities of assignment (±one SD) of each of the standards within an in silico mixture of all FCM standard datasets (subsampled to $n = 5000$ cells each, five 32-standard ANN classifiers). **b** Distributions of classification probabilities for four classes that were attributed in high numbers within the lake water community in the classifier results of **a** (i.e., B02, ACH2, CCR1 and PVR1) for each standard individually, for lake water (LW), or, in one case, of LW in silico mixed with $n = 5000$ cells of the PVR1 standard. Values within panels indicate the mean probability of the shown distribution, and correspond to the value plotted in **a**. **c** Mean class attribution (absolute cell numbers) of the lake water enriched community on 1-octanol ($n = 536,783$ cells), and of the pure culture isolate (OCT, $n = 63,824$ cells) derived from this enrichment grown on 1-octanol, both after three days of incubation, for one of the ANN-32 classifiers and for a new classifier that was trained using a dataset that in addition included FCM data from the OCT isolate itself (ANN-33). Numbers on the bars indicate the mean probability of class attribution. Image display calculations are detailed in "Supplementary Methods".

attributed cells from the lake water community (Fig. 4a, light gray bars, Supplementary Methods, Section 5.1–5.2). For most of the 32 classes, these mean probabilities were lower than those of the pure standards themselves (Fig. 4a, orange bars). For four

relatively abundant attributed classes in the lake water community (B02, ACH2, CCR1, and PVR1) we computed the probability distributions, which in all cases showed probabilities shifted to lower values compared to those of the pure standards (Fig. 4b,

Supplementary Methods, Section 5.3). The probability distributions may be used to calculate a classification similarity score. For example, the mean probability of predicted classification of cells in lake water to class B02 was 0.806, but that of the true standard B02 was 0.994, giving an average classification similarity of 81%. As discussed below, a ratio of this sort could form the basis of a similarity score between cells in an unknown microbiota and members of the standard set. Given that, except for the bead standards (e.g., B02), most strain standards have wider probability distributions (e.g., Fig. 4b), one could also consider a further form of thresholding or binning on the probability distributions to describe similarities of unknown cells to the standard categories. Importantly, this showed that the approach is versatile, so that cells in unknown microbiota can be attributed to standard classes, but their similarity to those classes can also be further analyzed.

To illustrate this effect of similarities further, we analyzed the probability distributions and classification similarity scores in the freshwater community enrichments and compared those to a strain that we isolated from the enrichment on 1-octanol (Fig. 4c, Supplementary Methods, Section 5.4). 16 S rRNA gene sequencing confirmed the isolate as a *Pseudomonas* sp.. The FCM signature of this pure culture was predominantly assigned by the 32-standard ANN classifiers to ATJ2 (0.891 mean probability of predicted classification, Fig. 4c, *OCT isolate*). With a new ANN classifier that was trained with a standard set that included the isolate itself in addition to the previous 32 (ANN-33 classifier), however, the cells were exclusively attributed to their own class (0.953 mean probability of predicted classification, Fig. 4c, Supplementary Methods, Section 5.5). The classification similarity score of the isolate to the attributed class in the ANN-32 classifier (ATJ2, Fig. 4c) was thus 0.891/0.953 = 93%. The new 33-standard ANN classifier confirmed this isolate to account for 15.8% of cells in the enrichment, which corresponds to the 19.5% of 16S rRNA amplicon sequences attributed to *Pseudomonas* (Fig. 3c).

Collectively, these experiments thus demonstrate that CellCognize can discriminate compositional shifts in an unknown microbial community, despite the relatively low number of (arbitrary) classes used here for the CellCognize pipeline (32 classes), and that mean probabilities of predicted classification or probability distributions can be further used to quantify similarities of cell attribution to the used classes.

## Discussion

We developed a supervised machine-learning ANN pipeline for FCM data, named CellCognize, to infer microbial "cell type" diversity in community samples from multiparametric FCM signatures of individual cells, by comparison to signatures of predefined strain and bead standards. ANNs were trained to differentiate multiparametric FCM data sets of five or 32 standards, resulting in classifiers that were subsequently used to predict class attribution of cell types of untrained microbial samples of known or unknown composition from their FCM data. For the learned standards, the pipeline was capable of differentiating among closely related strains, and even differentiating among cells within the same strain according to cell physiology, with respect to growth phase and prior culture medium. Our experiments with lake water microbial communities further attested to the ability of CellCognize to track compositional changes, for example during enrichment following chemical amendment, and to provide direct estimates of population growth and biomass yield. Finally, we show how mean probabilities of class assignment can be used to infer similarities to the used standards.

Our results were mainly obtained with a set of 32 standards, which were to some extent arbitrarily chosen. In a given application, the standard set should be targeted according to specific constraints or requirements of the intended microbiota samples in order to produce the highest class assignment probabilities. Our standards included eight types of beads, to cover a wide range of sizes (from 0.2 to 15 μm diameter), and 24 strains, to cover a range of microbial cell types and growth phases. Including beads has the advantage of capturing similarities to cell types in untrained samples for which no cultured standards are available (e.g., the attribution to B02 in lake water), and their large homogeneity. Although the overall precision and recall of the 32-standard ANN classifiers was high (79.2%), which is much larger than expected by chance (3% for a 32-standard classifier), there was substantial variation in the precision and recall rates, and the mean probability scores of class assignment. Indeed, whereas beads were differentiated with almost 99% accuracy, strain standards were more likely to be misclassified. The reasons that some strain standards were confused more than others was not due to cells belonging to the same taxonomic family, nor dependent on morphology (e.g., coccoid, rod), but more likely because of heterogeneity within the population of cells comprising the standard during ANN learning, and overall similarities in optical properties in the FCM channels employed. Previous studies have shown that closely related strains can be differentiated by their global multiparametric signature[27], as our initial PCA indicated. As another recent study has shown, global recognition may be optimized to differentiate individual cells in strain pairs, but projection to higher order mixtures lowers prediction accuracy substantially[28]. However, global differentiation involves attempting to cluster thousands of cells of a pure culture grown under standardized conditions, whereas CellCognize calculates the probability that an individual cell belongs to a predefined class. The recall of the ANN classifiers depends on the level of variation within a standard. Defining coherent standard (sub) populations from FCM data may thus require more optimized automated multidimensional algorithms[31]. In this respect, our ANN model, with only seven FCM input parameters (and only a single DNA staining using SYBR Green I) yielded classifiers with a mean accuracy of 80%, albeit with some misclassification among certain strain standards (e.g., LLC, PPT, PMG, SWT). We would expect that classification could be further improved by employing specific fluorescent dyes that yield additional independent cell characteristics[32].

The greatest advantage of CellCognize may lie in quantification of targeted microbial strains and their physiologies within recurring "known" microbiota, for instance, in clinical settings, animal husbandry or engineered applications. Importantly, the CellCognize pipeline is not limited to the set of 32 standards, which we deployed here as a general broad platform and proof of principle. Previous studies with marine algae that are easier to differentiate because of their larger size and autofluorescence, have targeted up to 70 species[26]. Strain standards can be optimized for any new target microbial community or subset of strains. As an example, by including enrichment isolates we showed how initial ANN classifiers can be further optimized for their target community. Staining procedures and selected standard strains can be optimized, and be well characterized beforehand to give greater confidence in the resulting assignments. Imaging cytometry may provide further advantages for cell resolution and differentiation[33]. Evidently, we observed some interexperiment variation on independently regrown and analyzed cultures of *E. coli, A. johnsonii* and *P. veronii* (Fig. 2) while still capturing the majority of target classes correctly. This may have resulted from small differences in growth conditions or handling and may be better controlled by using fixed samples and

standards. Instrument variation may be further controlled by including standard beads, whose FCM multidimensional signature can be used to 'reposition' and normalize the other data. As more experience is gained, additional isolates can be included in the set of standards and classifiers re-trained. Given the very rapid analysis time of FCM (ca. 5 min per sample), and almost instant classification once the ANN algorithm is developed, a quantitative cell type diversity or focal strain analysis becomes very simple and fast, with low reagent costs.

There remains the question to what extent CellCognize can estimate diversity changes in unknown samples. Clearly, with the mean probability scores of class assignment that we obtained here, even perfect recognition of strain standards is challenging. Furthermore, cell type diversity is not a priori the same as taxonomic diversity, although they may overlap when using appropriate reference standards. On the other hand, the ANN classifiers correctly captured broad shifts in the composition of a lake water microbial community upon amendment with phenol or 1-octanol. Furthermore, the cell type diversity inferred using CellCognize was largely comparable with class-level molecular diversity analyses from amplicon sequencing, and went significantly beyond the number of clusters identified by unsupervised clustering methods on similar data sets[22]. However, the current difficulty is to interpret probability scores from classification and to translate them into similarities between unknown cells and the predefined standards, given the different precision and recall of the various standards themselves. Classification of unknown microbiota based on their highest probability among the standard classes will result in cell types being attributed to the most likely class among the predefined standards. Alternatively, one could take the mean probability assignment to a class and score the difference or ratio with respect to that of the true standard, to derive a similarity score. Other authors have introduced cell distance measures as a way to indicate similarities in machine learning classification algorithms[34]. It will be important to better understand how cell type dissimilarities can be described by an appropriate scoring metric and relate to, for example, different physiological conditions or between strains.

Several recent studies have emphasized the importance to include population or community density measurements of microbiota in addition to relative abundances from molecular taxonomic assessments, as this determines to a large extent functional response of microbiota[8–10]. FCM quantifies total cell numbers in a microbiota sample[10], and, thus, the ability to further differentiate these counts into individual populations[28,30] or growth phases[35] within the microbiota would be extremely valuable. The CellCognize pipeline represents a promising tool to achieve this, as it provides the ability to detect and quantify focal strains, can detect different growth phases of focal strains even within a diverse microbiota background, can track the growth and enrichment of populations within a community, and can be deployed to estimate biomass. We acknowledge that biomass calculations should be further improved: they are very sensitive to the existing estimates of the mass of individual standards, which could be improved using recent techniques[36,37]. Nevertheless, the alternative $^{14}$C-methods carry their own disadvantages, tending to overestimate substrate usage as a result of unspecific sorption to cells.

In conclusion, we have presented CellCognize, a supervised machine-learning ANN-based pipeline to classify microbial cells and estimate output class population densities in microbiota from multidimensional FCM data. Detection and differentiation of cell types and specific strains in microbial samples could be further improved by exploiting the wide spectrum of general and specific fluorescent dyes[20]. The method can be tuned to the target microbiota by including strain standards derived from the target

itself, or can be used as a general cell type diversity method based on similarity scoring derived from assignment probabilities to a more general set of standards. Clearly, however, more data from different communities would have to be analyzed to qualify the potential and value of CellCognize and similar pipelines for more general diversity analysis. The low-cost, rapidity and ease of FCM quantitative single-cell analysis and fast downstream classification of cell populations makes this a powerful tool to expand and complement routine analysis of microbiota samples in a wide variety of areas including clinical settings.

## Methods

**Strain and bead standards**. The yeast and 14 bacterial pure cultures that were used as standards in building ANN classifiers are listed in Table 1. The strains were initially selected on the basis of their occurrence in aquatic systems, and with the aim of including variation in size and morphology (e.g., rod, coccus), as well as groups of strains with similar taxonomy (e.g., *Pseudomonas*, *Arthrobacter*, *Sphingomonas*). Strains were grown aseptically and individually in liquid media in biological triplicates until reaching stationary phase (conditions provided in Supplementary Table 1). For *E. coli*, we further included samples from exponential growth (OD$_{600}$ = 0.5) and from stationary phase (OD$_{600}$ = 2) on two different culture media (Supplementary Table 1). Culture samples were diluted in phosphate-buffered saline (PBS) to $10^5$ or $10^6$ cells ml$^{-1}$ and stained in 200 μl aliquots with 2 μl of diluted SYBR Green I solution (1:100 in dimethylsulfoxide; Molecular Probes) in the dark for 15–30 min at 20 °C for FCM analysis.

Bead standards consisted of polystyrene size calibration beads with diameters of 0.2, 0.5, 1, 2, 4, 6, 10, and 15 μm (Invitrogen), used in solution at concentrations of $1 \times 10^6$ (0.2 and 0.5 μm), $6 \times 10^7$ (1 μm), $3 \times 10^7$ (2 and 4 μm) and $2 \times 10^7$ (6, 10, and 15 μm) beads ml$^{-1}$. Beads were stored and prepared for FCM analysis according to the manufacturer's guidelines.

**Flow cytometric analysis**. For FCM analysis using a NovoCyte flow cytometer (ACEA Biosciences, Inc.), a total volume of 20 μl of stained sample was aspirated at 14 μl min$^{-1}$ at a sample acquisition rate of (maximally) 35,000 events s$^{-1}$. Samples were analyzed in two technical replicates. The NovoCyte cytometer has accurate volumetric-based cell counting hardware so that no calibration through addition of counting beads is necessary. The sheath flow rate was fixed at 6.5 ml min$^{-1}$, which corresponds to a core diameter of ~7.7 μm. The instrument threshold was set to 600 in the FITC-H channel (497 nm excitation and 520 ± 30 nm acquisition, to capture SYBR Green I fluorescence) and to 20 in the FSC-H channel for all samples in all experiments. Seven FCM parameters were recorded per particle (FITC-A, FITC-H, FSC-A, FSC-H, SSC-A, SSC-H and Width). Data sets were exported as. csv files and imported for filtering and artificial neural network analysis in MatLab (vs. 2017a, details are provided in Supplementary Methods).

**Data pretreatment**. FCM data were filtered for each of the seven parameters between a fixed lower (generally a value of 100) and an upper boundary ($10^5$–$10^7$), and then log$_{10}$-transformed (Supplementary Methods, Section 2). Filtered and log$_{10}$-transformed data for each of the standards were plotted in 2D-combinations of FITC-H, SSC-H and FSC-H to identify potential subpopulations (see, e.g., Supplementary Fig. 1). Subpopulations containing at least 5% of all data were gated and separated within the filtered data sets by setting lower and upper log-transformed boundaries in each of the three-parameter dimensions (i.e., FITC-H, SSC-H and FSC-H). This process resulted in a total of 32 standards: 8 bead and 24 strain data sets (see Table 1). For the preliminary experiment with three strains (see below), we used five standards (three strains, two of which had two subpopulations).

**Artificial neural network reconstruction**. The filtered and gated data sets of the standards ($n \sim 3 \times 10^5$ to $1.5 \times 10^6$ events per standard) were used as input for the development of ANN models (Supplementary Methods, Section 2.2). The datasets were randomly subsampled to 10,000 events per standard using the *datasample* function (Matlab v. R2017a). Crucially, the lower and upper boundary values imposed during the filtering process for each of the seven FCM parameters were added as two data points ("anchoring") per parameter to the first subsampled standard. This process of "anchoring" was sufficient to fix the multidimensional position of the data series for the subsequent machine-learning algorithm. Subsampled anchored datasets were concatenated and used as input into the ANN model, during which they were further scaled (between −1 and 1, hence the necessity to add the anchors) and randomly divided using *Dividerand* (Matlab v. R2017a) into three blocks: a training set (50% of the data), a validation set (25%) and a testing set (25%).

The ANN architecture consisted of a feed-forward back-propagation algorithm with one input, one hidden and one output layer. The input layer contained 7 nodes (corresponding to the 7 FCM parameters), whereas the output layer contained 5 (for the preliminary three-strain experiment) or 32 nodes (one for each of the standards in the full set). Input nodes were connected to the hidden layer by

the *sigmoid* function (Matlab v. 2017a), whereas the hidden layer nodes (20) were connected to the output by the *softmax* transfer function (Matlab v. 2017a). The input matrix was trained using the *trainscg* function (Matlab v. 2017a) in a 1000-cycle of training, validation and testing (performance goal = 0 | time = Inf | min grad = $10^{-6}$ | max fail = 6). Performance of the ANN was evaluated by crossentropy. The outcome of the ANN model is a classifier function, termed the ANN classifier, describing the correlations between input parameters and the five (proof-of-concept experiment, ANN-5) or 32 classes of the standard dataset (ANN-32). The process of subsampling, anchoring, concatenation and training was repeated five times independently on the full datasets, generating five slightly different ANN classifiers. The performance of the ANN classifiers is assessed on the basis of confusion matrices (Matlab v. 2017a), representing precision and recall rates for the complete in silico mixed set of standards, and the false prediction rate (as shown in Supplementary Fig. 2, Supplementary Dataset 1).

**CellCognize testing of unknown and standard-mixed communities.** In a first proof-of-concept experiment, we cultured *E. coli* MG1655, *P. veronii* and *A. johnsonii* individually in fivefold biological replicates to stationary phase, diluted cultures 1:1000 in PBS, and measured cells by FCM after staining with SYBR Green I either individually, or in different mixtures of all three strains combined. Individual and mixture data sets were analyzed with CellCognize using a set of five replicate ANN-5 classifiers, comparing expected added cell numbers of each of the three strains with their assigned class attributions from the ANN-5 classifiers (Supplementary Methods, Section 3.1–3.3).

An aquatic microbial community from Lake Geneva was recovered from 2 L of lake water, sampled at 1 m depth at a site close to the shore in Saint-Sulpice (46.517°N, 6.579°E), and used as an unknown background microbial community. Debris was removed by filtering the lake water through a nylon cell strainer with 40-μm pore size (Falcon, USA). Bacterial cells were then collected from the filtrate using a 0.2-μm pore size polyethersulfone membrane filter (Sartorius, Switzerland). The filter with the cells was resuspended during 2 h in artificial lake water mineral medium (ALW; containing, per L, 36.4 mg $CaCl_2 \cdot 2H_2O$, 0.25 mg $FeCl_3 \cdot 6H_2O$, 112.5 mg $MgSO_4 \cdot 7H_2O$, 43.5 mg $K_2HPO_4$, 17 mg $KH_2PO_4$, 33.4 mg $Na_2HPO_4 \cdot 2H_2O$, and 25 mg $NH_4NO_3$). Cell density in the ALW microbial suspension was then quantified and diluted to $10^5$ cells per ml. The diluted samples were stained with SYBR Green I for 30 min in the dark, and then measured in FCM, in three biological replicates, each with two technical replicates. FCM data were exported as .csv format, merged, filtered between lower and upper boundaries, and log-transformed for each of the seven FCM parameters as described above. The same two (low and high) anchor values per FCM parameter were then added to the dataset to ensure its proper 'positioning' during the ANN classifier computation. The lake water microbial community data were analyzed alone ($n = 5039$ cells), and also after being merged in silico with each of the 32 standards individually, randomly subsampled ($n = 5000$) for that purpose (Supplementary Methods, Section 3.6–3.9). Datasets were then classified using each of the five ANN-32 classifiers, in order to attribute all events to the predefined standard classes. In a further test, randomly subsampled FCM datasets ($n = 5000$) of three standards each (AJH1, MG_STAT_MM and PVR1) were merged in silico with the lake water community ($n = 5039$ cells) and reclassified using the ANN-32 classifiers. The recovery rate was calculated as the ratio of the number of cells from the standard attributed to its own class and the in silico added number. The mean probability and probability distribution of attribution were calculated for those particles assigned to each class (for example, in Fig. 4b, Supplementary Methods, Section 5.1–5.3).

In order to evaluate whether CellCognize could distinguish different cell physiologies, we classified FCM datasets of all four *E. coli* standards (representing different strains, culture media, and cell growth phases) individually (randomly subsampled to $n = 20,000$ cells) or as an in silico mixture with $n = 5000$ cells of each, using the five ANN-32 classifiers (Supplementary Methods, Section 3.5–3.7).

The performance of ANN-32 classifiers was further evaluated by mixing stationary phase-grown *E. coli* into the filtered lake water microbial community samples. *E. coli* MG1655 was cultured either on LB or on M9-CAA in biological triplicates. Cells were counted in stationary phase samples, diluted in artificial lake water, and added as $1.0 \times 10^4$ or $1.0 \times 10^5$ cells ml$^{-1}$ to the lake water community. Mixtures were stained and measured on FCM for comparison with lake water microbial community samples alone. Data were extracted, filtered, log-transformed and anchored as described above, and analyzed with the five ANN-32 classifiers for standard class attributions (Supplementary Methods, Section 3.10).

**Lake water microbial community enrichment.** In order to evaluate the ANN classification of an unknown community, we incubated the Lake Geneva water microbial community with either phenol or 1-octanol, or without any further amendment, for three days. Microorganisms were collected from 10 L Lake Geneva water by filtration (0.2–40 μm pore size) taken in November 2018, and re-suspended in 100 ml ALW in acid-treated closed 500-ml glass Schott flasks to obtain starting cell concentrations of $10^5$ cells ml$^{-1}$. Uniformly $^{14}C$-labeled phenol or 1-C $^{14}C$-labeled 1-octanol (ANAVA Trading SA) were dosed at 1000–5000 dpm ml$^{-1}$ in a mixture with unlabeled compound of the same type, to obtain total carbon concentrations of 0.1, 1 or 10 mg C l$^{-1}$. Incubations with unlabeled phenol were further repeated three times independently with Lake Geneva microbial

communities sampled in October and November 2017, and January 2019. Unamended inoculated ALW served as control for background growth, whereas amended but non-inoculated ALW served as abiotic controls. Triplicate flasks were prepared per assay, and incubated at 21 °C in the dark with 150 rpm rotary shaking. Aliquots of 1 ml were taken immediately after dosing the substrate, and then daily by syringes with needles without opening the caps, for cell staining with SYBR Green I and FCM analysis. FCM data were exported, filtered and anchored as described above, and used as input for ANN classification using the five ANN-32 classifiers (Supplementary Methods, Section 4.1–4.3).

A further 12 ml were sampled from each flask for $^{14}C$-analysis by needle and syringe without opening the caps. A subsample of 0.1 ml was taken to measure the radioactivity in aqueous solution. A 5-ml aliquot was filtered through 0.2-μm-pore size membrane filter to collect cell biomass, and a comparison subsample (0.1 ml) was taken from the filtrate. At day 3, the remaining solution after sampling (85 ml) was acidified to pH 3, $CO_2$ was purged from the liquid by air stripping during 1 h, and the solution was collected into three vials each containing 5 ml of 1 M NaOH. Vials were pooled and 0.5 ml was sampled. Aqueous samples or filtered cells were mixed in 5 ml liquid scintillation cocktail (Perkin Elmer) to measure the amount of $^{14}C$-CPM (counts per min), which was converted to DPM (disintegrations per min) by multiplying by a factor of 1/0.94 to correct for the instrument's efficiency. Mass balance values are reported in Supplementary Fig. 7.

**Community diversity analysis by 16S rRNA gene amplicon sequencing.** Lake Geneva water prokaryotic species diversity was determined by 16S rRNA gene amplicon sequencing. Triplicate samples from the enrichment experiment carried out with phenol and 1-octanol at 10 mg C l$^{-1}$ in January 2019 as described above were collected immediately after dosing substrate and after three days incubation at room temperature in the dark. Sample volumes were adjusted to have similar cell densities at both time points. Cells were collected on 0.2-μm membrane filters (PES, Sartorius) and stored in FastDNA Spin kit solution for soil (MPBio) at −80 °C until analysis. Prior to DNA extraction, we added two internal standards (with cell number adjusted to 1% of the total cell density measured in the sample by FCM) to normalize 16S rRNA gene sequence variant abundances across samples, if necessary. The internal standards were in-house honeybee gut microbiota isolates belonging to *Giliamella* and *Bifidobacterium* (kindly provided by Lucie Kesnerova, University of Lausanne), which are unlikely to be found in lake water. After DNA extraction according to the manufacturer's recommendations for the FastDNA Spin kit for soil (MPBio), the V3–V4 hypervariable region of the 16S rRNA gene was amplified using the 341f/785r primer set with appropriate Illumina adapters and barcodes. PCR conditions, amplifications and library preparations followed recommendations in the Illumina Amplicon sequencing protocol (https://support.illumina.com/documents/documentation/chemistry_documentation/16s/16s-metagenomic-library-prep-guide-15044223-b.pdf). Equal amounts of amplified DNA from each sample were pooled and sequenced bidirectionally on the Illumina MiSeq platform at the University of Lausanne. Raw 16 S rRNA gene amplicon sequences were separated by barcode, quality filtered, concatenated, verified for the absence of potential chimera, dereplicated and mapped to known bacterial classes (level 3) or species (level 7) using QIIME2 at 99% similarity to the SILVA taxonomic reference gene database on a UNIX platform[38]. Alpha- and beta-diversity measures were calculated in *R* using the *phyloseq* package. Significance of treatment and condition clustering was calculated using ADONIS implemented in *vegan* with the Bray–Curtis distance matrix and 999 permutations. Similarity of CellCognize and 16S rRNA amplicon (level 3) Bray–Curtis distance matrices was further assessed in a Spearman correlation of linear vectorized matrices (MatLab functions *squareform* "tovector" and *corr*, "type","Spearman"). Finally, the vectorized matrices were compared by MatLab *mdscale* and *procrustes* to calculate the goodness-of-fit dissimilarity measure *d*.

**Pure culture isolation.** Phenol- and 1-octanol-amended communities at day 3 were plated on MicroDish® platforms placed on silica gel disks with 10 mg C l$^{-1}$ of the same substrate and incubated for three days at 21 °C. Microcolonies were picked and transferred to glass vials with ALW and the same phenol or 1-octanol concentration for further propagation. One such isolate (named OCT in further analyses) was able to grow both with phenol and 1-octanol at 10 mg C l$^{-1}$ and was used for CellCognize classification as described in the *Results* section above. On the basis of its amplified and sequenced gene for 16S rRNA, this isolate identified with 99.5% nucleotide identity as *Pseudomonas azotoformans*. FCM data of a pure stained culture of the OCT-isolate grown in biological triplicates on ALW with 1-octanol for three days was included with the previous 32 standards to train a separate ANN-classifier (ANN-33), which was used to analyze the enrichment cultures (Supplementary Methods, Section 5.5).

**Estimation of microbial community biomass from CellCognize classification.** For the estimation of biomass using the ANN-32 classifiers, the mean number of classified events for each of the standard classes was multiplied by the estimated average C-mass per cell of that standard (Supplementary Table 2), and summed across all standard classes to obtain the total biomass at that time point in the community (as outlined in Fig. 1f).

To estimate the average C-mass per cell of the standards, biovolumes of each strain and bead standard individually in solution with a density of $10^7$ particles or cells ml$^{-1}$ was measured using a 3D cell explorer microscope (Nanolive). A 60× objective ($\lambda = 520$ nm), 0.2 mW mm$^{-2}$ light intensity was used for imaging with a resolution of $\Delta xy = 200$ nm, $\Delta z = 400$ nm, and a field of view of $85 \times 85 \times 30$ μm. At least five randomly selected fields were examined. Nanolive's STEWE software (https://nanolive.ch/software/) with the Image J plugin was deployed to segment particles on images and to calculate the average biovolume per cell per standard (Supplementary Fig. 3). The average standard biomass was then calculated from the corresponding biovolume using the allometric formula as proposed by ref. [39]

$$m_b = 435 \times V^{0.86} \qquad (1)$$

where $V$ represents the measured average biovolume (μm$^3$) and $m_b$ the calculated (dry weight) biomass (fg). Dry weight biomass values were divided by two to obtain the carbon mass per cell. Upper and lower boundaries were calculated in the same way, but by taking the mean biovolume plus or minus its measured standard deviation, respectively. Estimates of community biomass were compared with values obtained from $^{14}$C-substrate incorporation.

**Statistics and reproducibility**. Pure and mixed cultures, and aquatic samples were all grown in biological triplicates or in five-fold replicates. Different serial dilutions were prepared from every sample before staining and analysis in two technical replicates on flow cytometry, as is common for the field. At least between 10,000 and 100,000 cell events were recorded for all cultures and samples, if possible.

Events with negative data values from flow cytometry were excluded, and events were further thresholded to min- and max-values for each of the 7 measured FCM parameters (as specified in Supplementary Methods, section 1.1). Raw sequence reads were quality-controlled, cleaned and processed as described in the QIIME2 package, and potential chimera were removed.

Standard strains for the development of the ANN-32 classifiers and for the 3D imaging were cultured once in biological triplicates. Three of them (*A. johnsonii*, *E. coli* and *P. veronii*) were cultured on a separate occasion for the development of the ANN-5 classifiers (five biological replicates), and two others (*E. coli* MG1655 and *E. coli* DH5α-λpir) were cultured separately in biological replicates for mixing with the lake water bacteria. Lake water samples were taken, analysed or used for culturing on five different occasions as specified in the Methods sections above.

Cell data for inclusion in the development of the training and validation sets for the neural network, or for replicate testing, were randomly subsampled from the complete pool of data, as indicated in the main text and as described in the Supplementary Methods for each of the experiments.

The exact sample size ($n$) is given as a discrete number of cells or experimental replications, dependent on the experiment. Statistical parameters of central tendencies (e.g., means) or other basic estimates and variation (SD) are explained at appropriate positions in the main text and figure legends. Machine-learning outcomes (e.g., recall and precision, accuracy, receiver operating characteristics, probability scores, predicted classification) are explained in Supplementary Notes and Supplementary Methods, with details on the used scripts and formula. Mean and standard deviation on cell type classifications were calculated from the variation among the five independently generated ANN classifiers.

Principal component analysis of the variance among 32 standards for classification is explained in Supplementary Methods, Section 3.4. Comparison of diversity measures is described above. Biomass yield estimates were compared by two-sided t-test statistic on $^{14}$C-values from biological triplicates and biomass sums from predicted CellCognize classification on the same samples.

**Reporting summary**. Further information on research design is available in the Nature Research Reporting Summary linked to this article.

## Data availability
The raw data for the 16S rRNA amplicon sequencing data can be accessed from the BioProject PRJNA641590 using the accession code SAMN1535695-15356976. Flow cytometry and 16S rRNA amplicon sequencing data from this work are accessible from a single online accession at Zenodo.org (10.5281/zenodo.3822094)[40]. All source data are available as Supplementary Data in Excel format. Please see Description of Additional Supplementary Files for more information.

## Code availability
Detailed code and explanations is available from a single online accession at Zenodo.org (10.5281/zenodo.3822094)[40]. The detailed scripts for the CellCognize pipeline for every experiment and image display are provided in Supplementary Methods, with a table of content on p. 2 of the Supplementary Information.

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

## Acknowledgements
We thank Nora Khelidj for her help in the initial development and testing of the Cell-Cognize pipeline. Russell Naisbit, Philipp Engel and Martin Ackermann are thanked for their critical comments on the manuscript. This work was supported by grant 16800.1 PFIW-IW from the Swiss Commission for Technology and Innovation and by the National Centre in Competence Research in *Microbiomes*.

## Author contributions
B.D.Ö.D., A.F.B., and J.R.M. provided experimental data. B.D.Ö.D., N.H. and A.F.B. developed the initial scripting. B.D.Ö.D., N.H., and J.R.M. revised and improved the machine-learning scripts, and validated data comparisons. B.D.Ö.D., N.H., M.S., and J.R.M. analyzed data. B.D.Ö.D., N.H., and J.R.M. wrote the main text. All authors corrected and approved the final text.

## Competing interests
The authors declare no competing financial interests but the following competing non-financial interests: B.D.Ö.D. is the inventor on a patent application by the University of Lausanne that covers the CellCognize concept.
