## [Peer Review File · Communications Biology]

Reviewers' comments:

Reviewer #1 (Remarks to the Author):

Duygan and colleagues present a manuscript that builds upon a number of recently published papers that fully try to use the resolution of microbial flow cytometry (FCM). More specifically, the authors present CellCognize, a data analysis tool that consists of a trained artificial neural network (ANN) that uses flow cytometry to retrieve microbial population abundances by performing single-cell classification on in silico and synthetic microbial communities. Next to that, the authors tried to apply their method to retrieve shifts in abundances in a natural community as well.

They make three claims in the abstract:

- 1) The resulting classifiers were extensively validated in silico on known microbiota, showing on average 80% prediction accuracy.
- 2) Furthermore, the classifiers could detect shifts in microbial communities of unknown compositions upon chemical amendment, comparable to results from 16S-rRNA-amplicon analysis.
- 3) CellCognize was also able to quantify population growth and estimate total community biomass productivity, providing estimates similar to those from 14C-substrate incorporation.

While I find that the first one is sufficiently proven, the others, especially claim 2 and the connection to perform diversity analyses of unknown natural communities, are insufficiently laid out and demonstrated. The experimental setup concerning the latter is quite limited, and I find it difficult to understand the presented calculations and results. Moreover the connection to diversity is merely illustrated and hardly quantified. Therefore, the claims concerning the general ability of CellCognize to analyze natural communities are too strong. I think the authors should either considerably improve the Results section after "Analysis of diversity of unknown microbiota", including a proper presentation of the methods, quantify the results and potentially extend the experiment to truly prove that CellCognize is able to retrieve (some) properties of the microbial diversity of a natural community. Alternatively, as the paper is already quite loaded with results, the authors could shift the scope of the paper somewhat, focus more on the first part of the results section (claim 1) and tune down the second part.

Besides that, I have quite a number of questions and unclarities that I think the authors should consider. The manuscript contains a lot of experiments and results, for which some of them are not clearly explained, making this manuscript quite difficult to read. I will motivate these remarks in the rest of this document. I will use the numbers of references if they are already part of the manuscript, if not I will note the last name of the first author and add the full reference at the end of this document.

Introduction:

The introduction is written quite well, but I have a number of minor remarks:

- L65-66: I think two important papers are missing that related quite directly compositional changes in the microbial community (through 16S rRNA gene sequencing) and flow cytometry, see Garcia et al. (2015) and Props et al. (2016).
- Reference [28], at least in the introduction, should be replaced by Props et al. (2016).
- Note that the authors from [27] have evaluated the use of supervised machine learning methods up to in silico communities of 20 species, which is not so different from the number of evaluated bacterial species in this paper (which is 15 or 24 depending on how bacterial populations are defined).
- I distinguish three clear claims in the abstract that are clearly laid out. I suggest the authors have them better matched with the claims in the introduction (L79-87).

Materials & Methods:

- How did the authors determine the growth stage? Can the authors provide growth curve data (e.g. as supporting information) to demonstrate this?
- Probably the authors intend to say 'gating' by using 'anchoring' FCM. If so, please rephrase.
- Also, and this is very important, can the authors confirm that the defined gate to separate noise from data is the same for all samples? Otherwise, different gates per sample could induce a bias in the data that is picked up by the classifier, and this could be the reason of the high classification accuracy. If not, please define one gate for all the samples to separate background from sample data and rerun the analysis.
- Can the authors provide negative control samples to illustrate the choice of their gates?
- I am confused about the fact that the authors identified multiple subpopulations that belong to the same strain. What's the motivation to do this? If one class consists out of two or more phenotypic populations, why not classify these together? Doing this has the advantage of alleviating the step of manually inspecting and gating phenotypic populations.
- This is a detail, but important and often missed by researchers, but no information should be 'leaked' from the training to the test set. Although this step should not alter the results of the analysis too much, please perform the scaling to -1 and 1 using only the training set for this, i.e. use the mean and covariance of the training set to also rescale the test set.
- L464-466: What are the anticipated compositions of the mixtures?
- Is the gating the same for the analyzed natural community?
- I think calculating recovery rate is a very important and interesting approach but it should be better explained.
- Is it possible to provide a runnable script on for example Github or another code-sharing platform?
- As Communications Biology is an open-access journal, I suggest the authors make the raw data available as well. FlowRepository is a recommended data repository by Springer Nature to store FCM data (Spidlen et al., 2012).

Results:

- The authors propose the use of an ANN. Did the authors consider other models? For example Linear Discriminant Analysis (Abdelaal et al., 2019), Random Forests [27] or Support Vector Machines[16]?
- L102-103: Terminology is wrong here: You have a trained ANN model, that for example consists out of a set of learned linear equations, that based on the training data provides class assignments for unlabeled test data.
- L118: The ANN correctly assigned 76-88% of the cells in regrown pure cultures. So these regrown pure cultures are a separate test set? This is quite a convincing result, especially taking heterogeneous properties of microbial populations into account. Can you make this setup more explicit, also in the Materials & Methods section?
- L120: How were the expected proportions determined?
- L129-131: I don't understand how the PCA analysis was performed, this is also not included in the Materials & Methods section.
- Table 1 and other figures and tables when applicable: What's the total number of runs/samples? $n = 5$? Please note for each figure and table where applicable.
- While it's very useful to include beads in the training set, I would suggest to put the emphasis on the capacity to distinguish the 24 (or 15) strains.
- The beads could be very useful to further back-up the biovolume estimations, see for example the work of Haraguchi et al. (2017).
- L162-L163: What is meant with 'the experimental dataset' versus the 'in silico' mixed FCM dataset?
- L162-170: It is quite well known that FCM enables a differentiation of the same strain at different growth stages, see for example Melzer et al. (2015), this might be useful to add to the discussion.
- Fig. 3A: I suggest the authors use a logarithmic scale when describing absolute cell counts.
- L233-235: The authors state 'these results suggest that distinct cell types are reproducibly enriched

by carbon amendment, and that CellCognize is able to quantify such changes in community composition'. While I agree that CellCognize is able to detect the changes, at this stage there is no proof that the quantification is correct and makes sense (this can only be done via 16S rRNA gene sequencing).

- L239-241: Can you quantify this statement?

- L243: "Shannon diversity was moderately correlated between both methods". What is moderately? Which test is used? Is it significant, at which level?

- L244: "Both methods grouped replicates, treatments and time effect equally well": Can you quantify this? For example using a Mantel or Procrustes test?

- L245: "broad changes in communities can be captured equally well": I think again that both methods can detect community changes, and that it would be more useful that if 16S rRNA gene sequencing is considered as the golden standard (which of course has its own biases), the correlation with CellCognize estimations is given.

- L252: What's the sample size (n = ...)?

- L254-256: I think that part of the reason that the biovolume estimations of CellCognize are within reason is that although cells assigned to a certain standard, they can belong to an entirely different strain. However, they will have similar (forward) scatter properties, which is a proxy for cell size and biovolume.

- L267-302: This part is quite unclear, as a description concerning the calculation is missing from M&M and the result is, although interesting, very lean to conclude that "CellCognize can discriminate compositional shifts remarkably well even in an unknown microbial community" (L298-299). Therefore I think this requires an in-depth analysis to really use the assigned probabilities of unknown species in natural communities, especially to retrieve the abundance of certain species. However, taking the length of the manuscript into account, the authors should reevaluate which to include, and fully describe in the manuscript, and which parts to omit.

Discussion:

- 'standards': Please rephrase and make the distinction between beads and biological strains.

- I am missing a note on the reproducibility of the approach, especially with regard to the phenotypic heterogeneity of microbial populations.

- L336: Note that the authors from [27] also performed single-cell classification.

- Please try to use objective or quantifiable statements instead of e.g. 'very promising' (L343).

- L347-349: I agree with this statement, however, the part concerning the analysis of natural communities is less proven.

- L365-368: Please clarify and correct this statement, as the authors of works 21-23 do not cluster the data, but employ cytometric fingerprinting approaches to analyze their samples.

- Reference 31 feels somewhat disconnected to the rest of the manuscript in its current state.

- I would be very careful calling the estimations of CellCognize diversity for natural communities, as this is based on strains that are not necessarily part of the natural community. Therefore, to put emphasis on this in the title, or to say that CellCognize "can be used as a general diversity method" is too strong, and should either be more extensively proven or rephrased. In its current state I would conclude that CellCognize is able to detect shifts in the microbial community composition of natural (unknown) communities.

References:

García, F. C., Alonso-Sáez, L., Morán, X. A. G., and López-Urrutia, Á. (2015). Seasonality in molecular and cytometric diversity of marine bacterioplankton: The re-shuffling of bacterial taxa by vertical mixing. *Environmental Microbiology*, 17(10):4133–4142.

Haraguchi, L., Jakobsen, H. H., Lundholm, N., Carstensen, J. (2017). Monitoring natural phytoplankton communities: A comparison between traditional methods and pulse-shape recording flow cytometry.

Aquatic Microbial Ecology, 80(1):77-92.

Melzer S., Winter G., Jäger K., Hübschmann T., Hause G., Syrowatka F., Harms H., Tárnok A., Müller S (2015). Cytometric patterns reveal growth states of *Shewanella putrefaciens*. *Microbial biotechnology* 8(3):379-391.

Props, R., Monsieus, P., Mysara, M., Clement, L., and Boon, N. (2016). Measuring the biodiversity of microbial communities by flow cytometry. *Methods in Ecology and Evolution*, 7(11):1376–1385.

Spidlen, J., Breuer, K., Rosenberg, C., Kotecha, N., and Brinkman, R. R. (2012). FlowRepository: A resource of annotated flow cytometry datasets associated with peer-reviewed publications. *Cytometry Part A*, 81A(9):727–731.

Reviewer #2 (Remarks to the Author):

The manuscript has its background in the field of decontamination of toxic components in natural environments by using artificial or natural microbial communities instead of pure cultures. The authors were interested in the functioning of microbial communities under such conditions by determining their composition and estimating the degradation potential of auspicious members regarding their respective cell quantity increase and individual cell biomass growth.

The authors used flow cytometry towards this aim, a method which provides absolute cell numbers and allows for detection of community structure shifts within minutes after measurement. The main innovation of the study was the development of an artificial neuronal network tool called CellCognize which enables the automatic detection of microbial community diversity parameters such as richness and evenness using flow cytometric data and especially to estimate biomass of classified events in a (standardized) microbial community. I'm not familiar with machine learning but the pipeline for this endeavour was clearly and logically presented. A standard test dataset of monodisperse beads and known bacterial strains was used to train CellCognize. This standard was also tested before the background of a natural microbial community and finally, and most demanding, by using different strains of *E. coli* in different growth states. CellCognize can obviously also be trained further when a certain species develops into an important member of a community, thus creating a new CellCognize class. The success of the tool was verified by 16S rRNA gene amplicon sequencing and specific -14C -substrate incorporation (after degradation of toxic components).

CellCognize is indeed an outstanding tool because it follows basically cell growth and variations thereof between members of a microbial community which is the most important information on cell activity that can be obtained. This makes the approach unique at the time.

While it is obvious that members of low complex microbial communities might be easily recognized, even in their subpopulations, it would also be interesting to know the upper limit of cell classes that can be resolved by CellCognize. Would this be also dependent on the number and type of cell parameters that are used for classification? In this study basically only FSC, SSC, and one fluorescent parameter were used for the classification.

Another major question seems to be the cytometric pattern resolution capacity of the CellCognize tool which probably also relates to the question before. Single species change their cytometric patterns during their various states of growth, shifting their patterns into different locations of a 2D-plot. In addition, species such *Bacillus subtilis* can have various types and numbers of subpopulations

depending on growth states. Even if CellCognize can recognize both shifts and subpopulations, what happens if subpopulations overlap in certain activity states of a community? Is there a strategy available to calculate the different proportions of cell(sub)types in one classifier?

Furthermore, it would be interesting to learn if the classifier works equally well in evenly distributed communities in comparison to those that are dominated by only a few (sub)populations. How does CellCognize realize the borderline to a neighbouring class?

Minor questions:

How low can the abundance of a cell type be to still find it in a microbial community by the Classifier? Are there lower or upper limits?

Does it need always pure cultures to train the Classifier? Or can this also be done for natural communities? In this case, how can the results be proven?

Supplementary information: How many different cell types and growth states per strain have been included into the biovolume analysis standardization? Bacterial biovolume is hugely dependent on growth states and species can develop an awful number of morphologically different subpopulations. However, in S3 only a few cell types seem to have been tested.

Minor points:

L132-125: numbers of standards: 8 beads, 1 yeast, 14 bacterial populations + 8 subpopulations = 31

L 419 and following: Set up of the flow cytometer:

The measurement of 35,000 events per sec seems to be rather fast, better resolutions might be obtained at lower flow rates. Was this tested?

Please, remove the FITC-channel from the description: it is either green FI or SYBRGreen FI.

Please provide the gate that was obviously used to exclude background noise from the instrument, medium, fresh water or cell debris.

Please upload your raw cytometric data into the Flowrepository: Flowrepository.org/ and state if you handled the data according to the MiFlowCyt Rules:

<https://onlinelibrary.wiley.com/doi/pdf/10.1002/cyto.a.20941>

Please state, how many cells of a cytometric 2D-plot were included in the CellCognize training. Did you use always the same number of cells? Did you set a threshold to exclude low abundant cells or events?

Please provide a time which was necessary to classify a 32 artificial community dataset and, in addition, to classify a new upcoming community member? Is CellCognize able to instantly recognize newly upcoming cell clusters within otherwise constant community patterns? How long does it take to train on an unknown community data set? Or is it impossible to train CellCognize using not standardized information?

Figure 1b: Please exchange FITC channel against SYBRGreen channel

Figure 2b: Some of the strains/strain+conditions seem to be very similar in their classification. How

high is the overlap of these classes? Can this be used as an advantage?

Table S2: Its unclear if biovolume and biomass was calculated of only one cell type per strain or if the values origin from the average of different cell types per strain. How many cells were measured? Please explain.

Figure S1: Please remove the description FITC channel, see above

Dear Prof van der Meer,

Your manuscript entitled "Rapid detection of microbiota diversity using machine-learned classification of flow cytometry data" has now been seen by 2 referees. You will see from their comments below that while they find your work of interest, some important points are raised. We are interested in the possibility of publishing your study in *Communications Biology*, but would like to consider your response to these concerns in the form of a revised manuscript before we make a final decision on publication.

Reply: We thank you for the overall positive response and your willingness to consider our manuscript revision. See below for detailed comments.

Both reviewers thought the work is very useful and had some specific clarification points. R1 asked for quantification and some statistical tests, to make the raw data and code available, provide more details in the methods, and for some selectivity in what to be included. R2 mostly had clarifications about the applications for this method and asked to also make the data available.

Reply: We thank both reviewers for the detailed suggestions and questions. We have revised the manuscript accordingly and have added a step by step guideline in a new Supplementary Methods for reproducing all the figures in the manuscript, together with detailed technical clarifications, extra examples and required scripts. We believe that this new section should resolve all detailed methodological questions.

The script is (was) provided as general script, but is now detailed for the individual analyses of all Figure Display items in the Supplementary Methods.

FCM data are uploaded with the scripts to a single open access online source at Zenodo.

Reviewers' comments:

Reviewer #1 (Remarks to the Author):

Duygan and colleagues present a manuscript that builds upon a number of recently published papers that fully try to use the resolution of microbial flow cytometry (FCM). More specifically, the authors present CellCognize, a data analysis tool that consists of a trained artificial neural network (ANN) that uses flow cytometry to retrieve microbial population abundances by performing single-cell classification on in silico and synthetic microbial communities. Next to that, the authors tried to apply their method to retrieve shifts in abundances in a natural community as well.

They make three claims in the abstract:

- 1) The resulting classifiers were extensively validated in silico on known microbiota, showing on average 80% prediction accuracy.
- 2) Furthermore, the classifiers could detect shifts in microbial communities of unknown compositions upon chemical amendment, comparable to results from 16S-rRNA-amplicon analysis.
- 3) CellCognize was also able to quantify population growth and estimate total community biomass productivity, providing estimates similar to those from 14C-substrate incorporation.

While I find that the first one is sufficiently proven, the others, especially claim 2 and the connection to perform diversity analyses of unknown natural communities, are insufficiently laid out and demonstrated. The experimental setup concerning the latter is quite limited, and I find it difficult to understand the presented calculations and results. Moreover the connection to diversity is merely illustrated and hardly quantified. Therefore, the claims concerning the general ability of CellCognize

to analyze natural communities are too strong. I think the authors should either considerably improve the Results section after “Analysis of diversity of unknown microbiota”, including a proper presentation of the methods, quantify the results and potentially extend the experiment to truly prove that CellCognize is able to retrieve (some) properties of the microbial diversity of a natural community. Alternatively, as the paper is already quite loaded with results, the authors could shift the scope of the paper somewhat, focus more on the first part of the results section (claim 1) and tune down the second part.

Reply: We thank the reviewer for the correct summary of our main results, and the general positive statement about our work. We understand the concerns regarding the application of CellCognize for community diversity, which we have phrased ourselves openly. For example, we have consistently phrased our method as potentially useful for cell diversity and have mentioned that this is NOT taxonomic diversity (l. 102, 244, 382, 450).

Still, we believe that it was important to show the potential and limitations of the technique for general diversity measurements, and to provide an outlook of possible improvements. We feel that we have been very moderate not to overclaim anything here. For example, in the abstract (l. 30/31) we say that ‘the classifiers could detect shifts in microbial communities of unknown composition’ and in l. 36/36 we conclude that ‘the pipeline is particularly suitable for recurring microbiota types,..., for which cell recognition can be optimized’. See further lines 450-452 in the discussion. Diversity of a system, even a microbial one, is not a priori the same as taxonomic diversity.

As we further outline in the paragraph starting on l. 206, the most reasonable proof-of-concept for measuring relevant diversity changes in unknown microbiota (of note: we do not say ‘natural communities’, but ‘untrained’ or ‘unknown’) was to expose a community to low specific carbon substrate, because it would necessarily induce changes. We considered that a good test for CellCognize to demonstrate its capability to quantify such changes. We felt this was more relevant than testing diversity in a number of unrelated natural samples, because at this point we had no benchmark to compare diversity of natural samples. We believe this is a sound experimental strategy, which has been taken by other authors (e.g., inducing treatments in reactors with mixed microbial communities, and follow diversity changes both with FCM and 16S rRNA gene amplicon analysis).

With respect, but we believe that we have shown that CellCognize ‘is able to retrieve (some) properties’. The exposure experiment clearly showed that CellCognize is able to detect the enrichment. Diversity measures can be retrieved for both methods (CellCognize and 16S rRNA amplicon diversity), as we show in Fig. 3. Figure 3 is a relevant illustration and comparison of diversity measures, while acknowledging that CellCognize at this point only has a restricted number of classes that is much lower than 16S rRNA amplicon diversity.

Actions: We prefer at this point not to take out this aspect of CellCognize analysis, because it is a proof of principle that further work can improve on. We have, however, modified the text at relevant positions to not speak of ‘measuring general diversity’, but detecting shifts in community composition and specify ‘cell type diversity’ as opposed to ‘taxonomic diversity’.

In addition we have provided a Supplementary Method that details ALL technical aspects of the analysis with detailed script instructions, which we feel is merited by several comments of this and the second reviewer.

Finally, we have further quantified the diversity comparisons shown in Fig. 3, using a Mantel test on the distance matrices from CellCognize and 16S rRNA gene amplicon, a Spearman correlation and a procrustes test of the MDS plots. Furthermore, we tested for significance of replicate grouping in Fig.

3d using ADONIS. These values and description are included in the results on lines 246-252 and Supplementary methods, section 4.3

Besides that, I have quite a number of questions and unclarities that I think the authors should consider. The manuscript contains a lot of experiments and results, for which some of them are not clearly explained, making this manuscript quite difficult to read. I will motivate these remarks in the rest of this document. I will use the numbers of references if they are already part of the manuscript, if not I will note the last name of the first author and add the full reference at the end of this document.

Reply: We are very grateful for all the detailed remarks by this reviewer, showing the genuine interest in the work. We realize that in several descriptions we had to be succinct. In order to provide more background and details, we have added a Supplementary Method section for each of the experiments. This also includes the specific scripts for each of the analyses. In addition, we provide a 'glossary' of terms in the Supplementary Notes, to ensure that there is a little as possible confusion in terms.

Introduction:

The introduction is written quite well, but I have a number of minor remarks:

- L65-66: I think two important papers are missing that related quite directly compositional changes in the microbial community (through 16S rRNA gene sequencing) and flow cytometry, see Garcia et al. (2015) and Props et al. (2016).
- Reference [28], at least in the introduction, should be replaced by Props et al. (2016).
- Note that the authors from [27] have evaluated the use of supervised machine learning methods up to in silico communities of 20 species, which is not so different from the number of evaluated bacterial species in this paper (which is 15 or 24 depending on how bacterial populations are defined).

Reply and action: we have changed the references and text of line 71 accordingly.

- I distinguish three clear claims in the abstract that are clearly laid out. I suggest the authors have them better matched with the claims in the introduction (L79-87).

Reply and action: we verified the wording of our 'claims' to be as matching as possible (without wordly repeating).

Materials & Methods:

- How did the authors determine the growth stage? Can the authors provide growth curve data (e.g. as supporting information) to demonstrate this?

Reply: For most strains, this consisted of a standard incubation of 1 or 2 days, following by culture turbidity measurements. For E. coli, exponential growth was considered at turbidity of OD600 of 0.5, and stationary phase at OD=2.

Apart from E. coli, where we specifically included different growth phases, for the other species the selection of growth phase was arbitrary, because the only purpose here was to provide a cell/strain signature to which unknown cells could be compared to.

Action: this is specified now in line 411.

- Probably the authors intend to say 'gating' by using 'anchoring' FCM. If so, please rephrase.

Reply: We thank the reviewer for the remark and we realize that terminology is perhaps not crystal clear. No, the anchoring process is different. We 'filtered' the FCM data (see for example, Section 1.1. in the SI methods) to remove noise data below and above threshold levels. We then 'gated' the FCM data for the standard (sub)populations to identify their signatures as coherent as possible. Finally, for the CellCognize analysis, we added 'anchors' to every dataset (standards and unknowns), corresponding to the filter values (min and max). This is specified in Section 1.1 and 1.2 of the Supplementary Methods. The anchoring is crucial, because it keeps the dataset 'in place' during the scaling that is done during the neural network processing.

Action: we explained this more extensively in the Supplementary Methods, Sections 1.1 and 1.2, illustrating with the appropriate code. We have added a glossary of terms to avoid confusion as Supplementary Notes.

- Also, and this is very important, can the authors confirm that the defined gate to separate noise from data is the same for all samples? Otherwise, different gates per sample could induce a bias in the data that is picked up by the classifier, and this could be the reason of the high classification accuracy. If not, please define one gate for all the samples to separate background from sample data and rerun the analysis.

Reply: We explained above the process of filtering, gating and anchoring. This is illustrated now more extensively in the Supplementary Methods. All samples are analyzed in the same way.

Action: we explained this more extensively in the Supplementary Methods, Sections 1.1 and 1.2, illustrated with the appropriate code.

- Can the authors provide negative control samples to illustrate the choice of their gates?

Reply: There is no gating on individual samples. We only 'gate' to separate subpopulations for the development of the standards, which is justified and illustrated in Fig. S1 and in the sections 1.1 and 1.2 of the SI methods.

Action: we explained this more extensively in the Supplementary Methods, Sections 1.1 and 1.2, illustrating with the appropriate code.

- I am confused about the fact that that the authors identified multiple subpopulations that belong to the same strain. What's the motivation to do this? If one class consists out of two or more phenotypic populations, why not classify these together? Doing this has the advantage of alleviating the step of manually inspecting and gating phenotypic populations.

Reply: Subpopulations have a different FCM multidimensional characteristic. This is a specific signature that needs to be detected separately (as the E. coli example attests). If necessary, one can combine the two signatures to belonging to the same species when calculating total species numbers.

It is crucial to do it this way; otherwise the multidimensional signatures are not sufficiently coherent to be differentiated.

Action: we explained this more extensively in the Supplementary Methods, Sections 1.1 and 1.2, illustrating with the appropriate code.

- This is a detail, but important and often missed by researchers, but no information should be 'leaked' from the training to the test set. Although this step should not alter the results of the analysis too much, please perform the scaling to -1 and 1 using only the training set for this, i.e. use the mean and covariance of the training set to also rescale the test set.

Reply: Scaling of the data is from -1 and 1. At that point the MatLab routine randomly subsamples to a training, validation and test set. There is no overlap between data points.

Action: we explained this more extensively in the Supplementary Methods, Sections 2.1 and 2.2, illustrated with the appropriate code.

- L464-466: What are the anticipated compositions of the mixtures?

Reply: Anticipated compositions were calculated from the mixed volumes of the individual cell suspensions, which were quantified by FCM separately.

Action: This is explained and illustrated with code in Sections 3.1, 3.2 and 3.3 of the Supplementary Methods. We further refer to the glossary of terms in the Supplementary Notes: 'Predicted classification' and 'Correct predicted classification'.

- Is the gating the same for the analyzed natural community?

Reply: Filtering and anchoring yes. There is no gating for the lake water microbiota or any of the unknown microbiota samples.

Action: we explained this more extensively in the Supplementary Methods, Sections 3.7 and 3.8, illustrating with the appropriate code.

- I think calculating recovery rate is a very important and interesting approach but it should be better explained.

Reply and action: We thank the reviewer for the remark. Calculation recovery is further detailed in Section 3.3 of the SI methods with an appropriate example. We further refer to the glossary for terms: 'Predicted classification' and 'Correct predicted classification'.

- Is it possible to provide a runnable script on for example Github or another code-sharing platform?

Reply: We already provided the full script as SI data to the manuscript. The 'problem' with scripts is that, preferably, one provides a generalized script, but this is not immediately runnable because file locations and names have to be indicated. A runnable script would also need example files to be in the same location.

Action: We have provided real runnable code directly in the Supplementary Methods with appropriate examples. This can be simply copied and pasted in MatLab. We also provide the scripts with the different datasets as importable format on Zenodo.

- As Communications Biology is an open-access journal, I suggest the authors make the raw data available as well. FlowRepository is a recommended data repository by Springer Nature to store FCM data (Spidlen et al., 2012).

All data and scripts are available from a single DOI link at Zenodo.org (10.5281/zenodo.3822094)

Results:

- The authors propose the use of an ANN. Did the authors consider other models? For example Linear Discriminant Analysis (Abdelaal et al., 2019), Random Forests [27] or Support Vector Machines[16]?

Reply: We appreciate the suggestion of the reviewer. Based on suggestions of friendly neighboring experts in the area, we concentrated on the ANN network outline and for now did not consider running alternative models. None of the other methods mentioned address the “deep learning” as much as NNs do and with this type of data (both number of data points that are in the order of millions and relatively high dimensionality) we definitely needed a deep learning algorithm. Developing the proper running codes and analyzing data is very cumbersome. The community will be free to utilize the raw data for developing alternative models for strain recognition.
Action: No further action.

- L102-103: Terminology is wrong here: You have a trained ANN model, that for example consists out of a set of learned linear equations, that based on the training data provides class assignments for unlabeled test data.

Reply: The reviewer is correct.

Action: We have verified our terminology and the phrase. See further the new provided glossary in Supplementary Notes.

- L118: The ANN correctly assigned 76-88% of the cells in regrown pure cultures. So these regrown pure cultures are a separate test set? This is quite a convincing result, especially taking heterogeneous properties of microbial populations into account. Can you make this setup more explicit, also in the Materials & Methods section?

Reply: Yes, the regrown cultures are a separate test set. We have explained the setup and the analysis in detail in Sections 3.1-3.3 of the Supplementary Methods.

- L120: How were the expected proportions determined?

Reply: We have explained the setup and the analysis in detail in Sections 3.1-3.3 of the Supplementary Methods.

- L129-131: I don't understand how the PCA analysis was performed, this is also not included in the Materials & Methods section.

Reply: We have explained the PCA analysis with the corresponding code in more detail in Sections 3.4 and 3.5 of the Supplementary Methods. The terminology was refined using the glossary in Supplementary Notes.

- Table 1 and other figures and tables when applicable: What's the total number of runs/samples? $n = 5$? Please note for each figure and table where applicable.

Reply: As we describe in the materials and methods section, we used two technical replicates in addition to biological replicates. This is specified in all legends. We prefer reserving the ' $n = xxx$ ' notation for the number of analyzed cells, in order to avoid confusion.

Action: Replicate numbers, and data sets specified in Supplementary Methods.

- While it's very useful to include beads in the training set, I would suggest to put the emphasis on the capacity to distinguish the 24 (or 15) strains.

Reply: We appreciate the remark by the reviewer. However, the beads are very useful for smaller cells in natural community for which we have no other 'standard' signature. As noted in Figure 2a or 4a, a significant proportion of natural lake water bacteria is assigned to the bead standard B02.

Action: We comment on the usefulness of including beads as standard in the Discussion.

- The beads could be very useful to further back-up the biovolume estimations, see for example the work of Haraguchi et al. (2017).

Reply: We appreciate the remark by the reviewer.

Action: We included this reference on line 536.

- L162-L163: What is meant with 'the experimental dataset' versus the 'in silico' mixed FCM dataset?

Reply: We have explained the setup and the analysis in detail in Sections 3.1-3.3 of the SI Methods.

- L162-170: It is quite well known that FCM enables a differentiation of the same strain at different growth stages, see for example Melzer et al. (2015), this might be useful to add to the discussion.

Reply: We appreciate the remark by the reviewer.

Action: We mention and included this reference on line 505.

- Fig. 3A: I suggest the authors use a logarithmic scale when describing absolute cell counts.

Reply: We appreciate the remark by the reviewer, but prefer here to emphasize the magnitude of growth increase. The normalized values are shown in Fig. 3B. Log scaled values are shown in Fig. S5.

Action: no further action.

- L233-235: The authors state 'these results suggest that distinct cell types are reproducibly enriched by carbon amendment, and that CellCognize is able to quantify such changes in community composition'. While I agree that CellCognize is able to detect the changes, at this stage there is no proof that the quantification is correct and makes sense (this can only be done via 16S rRNA gene sequencing).

Reply: We appreciate the remark by the reviewer. The quantification is compared to 16S rRNA gene sequencing as is described in the paragraph 305-328.

Action: no further action.

- L239-241: Can you quantify this statement?

Reply: We appreciate the suggestion, but think that with normalized plots in percentages, it is relatively easy to see an enrichment of ~80% of the cells/taxa in both graphs of Fig. 3c. A more precise quantification is described for one of the isolates (L. 375-392).

Action: no further action needed.

- L243: "Shannon diversity was moderately correlated between both methods". What is moderately? Which test is used? Is it significant, at which level?

Reply: We appreciate this concern. The correlation of the Shannon diversity measures is presented in Fig. 3d ($r^2=0.5767$). This is a 'moderate' correlation. No further testing is needed for a correlation coefficient.

Action: No further action needed.

- L244: "Both methods grouped replicates, treatments and time effect equally well": Can you quantify this? For example using a Mantel or Procrustes test?

Reply: We appreciate the concern and question. We quantified the replicate and treatment grouping by both methods using ADONIS in the R-vegan package for Bray-Curtis distances of CellCognize or 16S rRNA amplicon sequencing. Both are smaller than $p=0.001$, and, therefore describe replicate and treatment clustering equally well. Distance matrices were further directly compared using Spearman correlations ($r=0.9005$, $p = 0$) and using procrustes on the ordinate plots ($d=0.2144$). This indicates they are similar.

Action: These descriptions were included in the results section (L. 323-326) and in Materials and methods (l. 687-692)

- L245: “broad changes in communities can be captured equally well”: I think again that both methods can detect community changes, and that it would be more useful that if 16S rRNA gene sequencing is considered as the golden standard (which of course has its own biases), the correlation with CellCognize estimations is given.

Reply: We appreciate to comment to clarify this. What we mean is that the underlying diversity measures are different (i.e., cell types and taxonomy).

Action: This is specified in l. 326.

- L252: What’s the sample size (n n = ...)?

Reply: As specified in the legend to Fig. S7 and in l. 516 of the Materials and Methods, these are biological replicate assays.

Action: We specified this again in l. 334.

- L254-256: I think that part of the reason that the biovolume estimations of CellCognize are within reason is that although cells assigned to a certain standard, they can belong to an entirely different strain. However, they will have similar (forward) scatter properties, which is a proxy for cell size and biovolume.

Reply: The reviewer is correct. However, lake water also has a considerable number of very small cells (falling in the B02 category, e.g. Fig. 2e), and, generally speaking, bacterial (buoyant) masses can be easily two logs different (Ref.32). Therefore, although a rough calculation with a single mass value may give the same result, it still makes sense to specify this better from the mass of individual standards.

Action: No further action.

- L267-302: This part is quite unclear, as a description concerning the calculation is missing from M&M and the result is, although interesting, very lean to conclude that “CellCognize can discriminate compositional shifts remarkably well even in an unknown microbial community” (L298-299). Therefore I think this requires an in-depth analysis to really use the assigned probabilities of unknown species in natural communities, especially to retrieve the the abundance of certain species. However, taking the length of the manuscript into account, the authors should reevaluate which to include, and fully describe in the manuscript, and which parts to omit.

Reply: we feel that the probability scoring and possibility for similarity calculation is an important part for readers to understand. Predictions of cell similarity come with a probability scoring that can tell something about the closeness of the cell type and the standard.

Action: we have included a more extensive methodological presentation in Section 5 of the SI Methods.

Discussion:

- 'standards': Please rephrase and make the distinction between beads and biological strains.

Reply: We appreciate the remark, but for the consistency of the manuscript, it is important to stick to one term. 'Standard' refers to the coherent FCM signature of a pure culture or bead suspension that is used in neural network analysis. As we specify again in l. 403, this consists of both beads and strains. See further the discussion in l. 427-440.

Action: no further action.

- I am missing a note on the reproducibility of the approach, especially with regard to the phenotypic heterogeneity of microbial populations.

Reply and action: We thank the reviewer for this suggestion and have included a paragraph in the discussion on this from line 476-481.

- L336: Note that the authors from [27] also performed single-cell classification.

Reply and action: We thank the reviewer for this indication, which we specified in line 444.

- Please try to use objective or quantifiable statements instead of e.g. 'very promising' (L343).

Reply and action: We thank the reviewer for this suggestion. We have rephrased this to 'with a mean accuracy of 80%' (l. 462 in the revised manuscript).

- L347-349: I agree with this statement, however, the part concerning the analysis of natural communities is less proven.

Reply: We thank the reviewer for this suggestion. The discussion of the 'less proven' part is presented in line 488-498. We believe we have been very moderate and careful in this discussion.

- L365-368: Please clarify and correct this statement, as the authors of works 21-23 do not cluster the data, but employ cytometric fingerprinting approaches to analyze their samples.

Reply and action: We thank the reviewer for this suggestion. Dhoble et al use machine learned classification. The others used FCM fingerprinting approaches.

Action: We have corrected the references here.

- Reference 31 feels somewhat disconnected to the rest of the manuscript in its current state.

Reply and action: We thank the reviewer for the suggestion and have rewritten this part from l. 502-506.

- I would be very careful calling the estimations of CellCognize diversity for natural communities, as this is based on strains that are not necessarily part of the natural community. Therefore, to put emphasis on this in the title, or to say that CellCognize "can be used as a general diversity method" is too strong, and should either be more extensively proven or rephrased. In its current state I would conclude that CellCognize is able to detect shifts in the microbial community composition of natural (unknown) communities.

Reply: We appreciate the remark by the reviewer. We have added another sentence to put the current state of affairs into perspective and 'tone down' the potential for general diversity analysis, while maintaining that diversity is not a priori the same as taxonomic diversity, and should not be reduced to that, either.

We feel the title is not wrong. We don't suggest a general biodiversity method or a replacement of taxonomic diversity.

Action: See line 539-541.

Reviewer #2 (Remarks to the Author):

The manuscript has its background in the field of decontamination of toxic components in natural environments by using artificial or natural microbial communities instead of pure cultures. The authors were interested in the functioning of microbial communities under such conditions by determining their composition and estimating the degradation potential of auspicious members regarding their respective cell quantity increase and individual cell biomass growth. The authors used flow cytometry towards this aim, a method which provides absolute cell numbers and allows for detection of community structure shifts within minutes after measurement. The main innovation of the study was the development of an artificial neuronal network tool called CellCognize which enables the automatic detection of microbial community diversity parameters such as richness and evenness using flow cytometric data and especially to estimate biomass of classified events in a (standardized) microbial community. I'm not familiar with machine learning but the pipeline for this endeavour was clearly and logically presented. A standard test dataset of monodisperse beads and known bacterial strains was used to train CellCognize. This standard was also tested before the background of a natural microbial community and finally, and most demanding, by using different strains of E. coli in different growth states. CellCognize can obviously also be trained further when a certain species develops into an important member of a community, thus creating a new CellCognize class. The success of the tool was verified by 16S rRNA gene amplicon sequencing and specific ^{14}C -substrate incorporation (after degradation of toxic components).

Reply: We thank the reviewer for the correct summary and the overall positive statement about our work.

CellCognize is indeed an outstanding tool because it follows basically cell growth and variations thereof between members of a microbial community which is the most important information on cell activity that can be obtained. This makes the approach unique at the time.

Reply: Thank you!

While it is obvious that members of low complex microbial communities might be easily recognized, even in their subpopulations, it would also be interesting to know the upper limit of cell classes that can be resolved by CellCognize. Would this be also dependent on the number and type of cell parameters that are used for classification? In this study basically only FSC, SSC, and one fluorescent parameter were used for the classification.

Reply: We thank the reviewer for this suggestion, which is, obviously, a bit difficult to answer given that this is the first study of this kind. Given that we find that some standards cannot be very well resolved whereas others can (e.g., Table 1), we think that expanding to more FCM parameters will be

necessary. We mention this in the discussion. It is like doing face recognition with only nose and eyes, but not the rest of the face. However, it will also be necessary to refine the procedures to define coherent multidimensional subpopulations of standards, which will require further statistical tools.

Action: We specified future expansion and optimization to more classes in the discussion on line 444-458.

Another major question seems to be the cytometric pattern resolution capacity of the CellCognize tool which probably also relates to the question before. Single species change their cytometric patterns during their various states of growth, shifting their patterns into different locations of a 2D-plot. In addition, species such *Bacillus subtilis* can have various types and numbers of subpopulations depending on growth states. Even if CellCognize can recognize both shifts and subpopulations, what happens if subpopulations overlap in certain activity states of a community? Is there a strategy available to calculate the different proportions of cell(sub)types in one classifier?

Reply: We thank the reviewer for the remark, which is again a very relevant question. Theoretically, we would assume that if cells have a different phenotype, this must show in cellular parameters. For example, exponentially growing cells are longer and have more DNA than stationary phase cells, which can be detected in FCM. Spores are smaller, so can be detected as well. Expression of different or heterogenous surface markers in subpopulations of cells can be detected by coloration with specific antibodies, etc. The optimization of CellCognize will thus depend on the specific question in mind, because it may require development and testing of staining methods. To count different subpopulations of the same standard is simply an a posteriori step, like in Fig. 2a. One can add the population proportions that belong to the same species.

Action: we explain the subpopulation addition in Section 3.1-3.3 of the Supplementary Methods. The question of experimental repeatability is now presented in l. 450-456.

Furthermore, it would be interesting to learn if the classifier works equally well in evenly distributed communities in comparison to those that are dominated by only a few (sub)populations. How does CellCognize realize the borderline to a neighbouring class?

Reply: We thank the reviewer for this suggestion. We tried to address this question with the synthetic community experiment by mixing equal proportions of strains, or one dominant strain (Fig. 2a). We further estimate this in the lake water enrichment experiment, comparing the native state and the enrichments over time and with different carbon sources (Fig. 3a). These data indicated that CellCognize detects relevant shifts in communities. On the other hand, very small proportions of focal strains are detected but the error increases proportionally (Fig. 2f). Class attributions can be estimated with the mean probability scores or by probability distributions (Fig. 4). One would need to gain more experience to better understand what this means exactly in terms of detecting the 'borders' between two strains.

Action: We explained the probability and similarity scoring more in detail in Section 5 of the new SI methods.

Minor questions:

How low can the abundance of a cell type be to still find it in a microbial community by the Classifier? Are there lower or upper limits?

Reply: We thank the reviewer for this question. The classifier will take each and every cell into account and makes a prediction of its assignment (as shown in the example of Dataset S2). There is therefore no upper or lower limit. However, it is the probability of the assignment that makes a difference. A single passing cell might be assigned with 99% probability to a single output class, but the false positive prediction rate is never zero (see the ROC curves in Fig. S2b).

Action: We explained the probability and similarity scoring more in detail in Section 5 of the new SI methods.

Does it need always pure cultures to train the Classifier? Or can this also be done for natural communities? In this case, how can the results be proven?

Reply: We thank the reviewer for this question. In absence of pure cultures, one has to rely on clustering or fingerprinting approaches to find coherent groups that can be gated and presented as a 'standard' to train the classifier. To understand the nature of those clusters will be very difficult, though, because one would have to separate them and characterize them by means of e.g., sequencing.

Action: No further action needed.

Supplementary information: How many different cell types and growth states per strain have been included into the biovolume analysis standardization? Bacterial biovolume is hugely dependent on growth states and species can develop an awful number of morphologically different subpopulations. However, in S3 only a few cell types seem to have been tested.

Reply: Holographic microscopy biovolume measurements were based on 5 images each per culture or bead suspension, containing each 25-50 objects. Due to the limited number of objects analyzed, no further subpopulations could be distinguished.

Action: This was specified in the footnote to Table S2.

Minor points:

L132-125: numbers of standards: 8 beads, 1 yeast, 14 bacterial populations + 8 subpopulations = 31

Reply: One of the bacterial standards has three subpopulations. See line 141

L 419 and following: Set up of the flow cytometer:

The measurement of 35,000 events per sec seems to be rather fast, better resolutions might be obtained at lower flow rates. Was this tested?

Reply: 35,000 per sec is the MAXIMUM rate. In most cases, the actual cell concentration is much lower ($10^5 - 10^6$ per ml, as specified in l. 532), which, at 14 μ l per min gives only 25-250 cells per sec.

Action: we specified the word 'maximum' to the acquisition rate.

Please, remove the FITC-channel from the description: it is either green FI or SYBRGreen FI.

Reply: We appreciate the suggestion, but would prefer to stick to the instrument channel (which is FITC-H or FITC-A). However, we specified that this channel captures SYBR Green I fluorescence in this case.

Action: Specified in l. 545-546

Please provide the gate that was obviously used to exclude background noise from the instrument, medium, fresh water or cell debris.

Reply: We appreciate the remark and apologize for confusion. The instrument threshold was specified in l. 545-546.

The filtering, anchoring and gating processes are different. We 'filtered' the FCM data (see for example, Section 1.1. in the SI methods) to remove outlier data below and above threshold levels. We then 'gated' the FCM data for the standard (sub)populations to identify their signatures as coherent as possible. Finally, for the CellCognize analysis, we added 'anchors' to every dataset (standards and unknowns), corresponding to the filter values (min and max).

Action: We specify the process in more detail in Section 1.1 and 1.2 of the Supplementary Methods. The anchoring is crucial, because it keeps the dataset 'in place' during the scaling that is done during the neural network processing.

Please upload your raw cytometric data into the Flowrepository: [Flowrepository.org/](https://flowrepository.org/) and state if you handled the data according to the MiFlowCyt

Rules: <https://onlinelibrary.wiley.com/doi/pdf/10.1002/cyto.a.20941>

Reply: We have submitted all raw data and scripts to a single online repository at Zenodo, for ease of using and understanding the ANN analysis.

Please state, how many cells of a cytometric 2D-plot were included in the CellCognize training. Did you use always the same number of cells? Did you set a threshold to exclude low abundant cells or events?

Reply and action: We apologize for the information density. All cell numbers and replicates are specified on every figure display or legend, and further in the Supplementary Methods.

Please provide a time which was necessary to classify a 32 artificial community dataset and, in addition, to classify a new upcoming community member? Is CellCognize able to instantly recognize newly upcoming cell clusters within otherwise constant community patterns? How long does it take to train on an unknown community data set? Or is it impossible to train CellCognize using not standardized information?

Reply and action: We thank the reviewer for the question. The analysis time is specified in l. 458-459. Given appropriate software development, it is probably possible to classify cells in real-time to an established classifier. This, however, is beyond what can be expected from this work. As we explain in the Supplementary Methods, one has to define the appropriate standards and provide this dataset for training, validation and testing to produce the classifiers. Once classifiers are obtained (the linear equations), subsequent classification is a matter of seconds.

Figure 1b: Please exchange FITC channel against SYBRGreen channel

Reply and action: see comment above on FITC channel. We specify in the legend that this is SYBR Green I fluorescence captured in the FITC channel (l. 847-848)

Figure 2b: Some of the strains/strain+conditions seem to be very similar in their classification. How high is the overlap of these classes? Can this be used as an advantage?

Reply: We thank the reviewer for the question. The 'overlap' is calculated as the precision of classification and is presented in Table 1.

Action: We specify the use of terminology in Supplementary Notes and explain the terms precision and recall in Supplementary Methods.

Table S2: Its unclear if biovolume and biomass was calculated of only one cell type per strain or if the values origin from the average of different cell types per strain. How many cells were measured? Please explain.

Reply: We thank the reviewer for the question. Biovolumes were calculated from 5 images per strain per culture containing an average of 25-50 objects. No subpopulations were separated from images.

Action. This was specified in Table S2 footnote.

Figure S1: Please remove the description FITC channel, see above

Reply and action: We specified in the legend to Figure S1 that the fluorescence in the FITC channel originates from SYBR Green I staining.

REVIEWERS' COMMENTS:

Reviewer #1 (Remarks to the Author):

Dear authors,

Dear editor,

I appreciate the effort of the authors to address the concerns I raised in previous rebuttal. These changes (such as the delineation between cell type identification and microbial diversity, nuancing and objectifying some of the results) have improved the manuscript.

At the same time, some (of the same) concerns remain. These include:

- The authors have provided an additional document supporting the manuscript. In its current state, this is a collection of additional experimental information and code, with no guidance in the manuscript on where to look in the supporting information file. If the Supporting Methods file is 20 pages long, it is insufficient to refer in the manuscript to the document as such. In addition, the response is often something of the like we have added this information to section X of SI. The quality of the SI file is not the same as that of the original manuscript. I think it is the challenge of the authors to present their method and results in such a way that the reader can easily grasp the presented materials, methods and results section, finding a balance between what should be in the main document and what is presented in the supporting file.

- The section "similarity assessment ... CellCognize classes" (line 320-377) remains unclear. What do the changes in probability distribution imply? What does the classification similarity mean? What do these values indicate? I tried to look this up in the Supplementary Methods, but still find the information insufficient.

- Please check the used machine learning terminology in the manuscript. For example:

- An ANN is constructed by a set of linear equations, but is a single classifier.

- 'Predicted classification': either you predict or classify the label (cell type in this case) of unknown instances, but 'predicted classification' does not make sense.

- 'machine-learned ...' is not the way to express that a method belongs to the field of machine learning. For example, you perform 'supervised classification' or you use a 'machine learning algorithm'.

- L81: None of these references (16,27,28,30) perform clustering. Reference 16 and 28 perform single-cell classification, in the same way as CellCognize. Reference 27 performs cytometric fingerprinting, and clusters the fingerprints at the community-level. Reference 30 calculates total cell counts, and proposes this as a methodology to go to absolute abundances by multiplying this number with taxon abundances.

- I tried to access the Zenodo repository, but I was not able to.

Other minor remarks:

Thanks for clarifying the intended use of cell type diversity. Maybe it's a good idea to also include this in the title?

I am not sure, or maybe I am misunderstanding the experimental setup, whether the experiment described in lines 209-225 is the correct setup. Standards are added to an unknown background of cells, but, as you perform single-cell classification, it makes sense that these standards are recognized, as they are classified independently from other cells. In case I am misunderstanding,

could you please improve the description of the experiment?

Reviewer #2 (Remarks to the Author):

The authors answered the questions sufficiently enough. I have no further questions.

Reviewer 1 asked to improve the presentation and readability of the methods in the supplementary material and its link to the main text, clarify some method sections, improve machine learning terminology, correct some references, correct the link for the zenodo repository, improve title and descriptions of the experiments.

Congratulations on an excellent paper!

Reply: We thank you for your appreciation of our work. Detailed comments are given below.

REVIEWERS' COMMENTS:

Reviewer #1 (Remarks to the Author):

Dear authors,

Dear editor,

I appreciate the effort of the authors to address the concerns I raised in previous rebuttal. These changes (such as the delineation between cell type identification and microbial diversity, nuancing and objectifying some of the results) have improved the manuscript.

At the same time, some (of the same) concerns remain. These include:

- The authors have provided an additional document supporting the manuscript. In its current state, this is a collection of additional experimental information and code, with no guidance in the manuscript on where to look in the supporting information file. If the Supporting Methods file is 20 pages long, it is insufficient to refer in the manuscript to the document as such. In addition, the response is often something of the like we have added this information to section X of SI. The quality of the SI file is not the same as that of the original manuscript. I think it is the challenge of the authors to present their method and results in such a way that the reader can easily grasp the presented materials, methods and results section, finding a balance between what should be in the main document and what is presented in the supporting file.

Reply: We concur that linking all data and descriptions remains complex, but we have done our best to make this as transparent as possible. The SI document on page 2 has an index with all paragraphs of the supplementary methods. This specifies per Figure panel how the analysis was done (which we explained in the main text under 'Script'). The source data are provided in Excel format. Therefore, even though the work is complex, this should enable to follow the various experiments. The reviewer thus cannot claim that there is 'no guidance in the manuscript on where to look'. The SI methods specify the strategy, the code, the files, the outcome of the experiment and give other examples.

Our previous reply that 'we have added this information to section X of SI' was exactly what was asked in the first round of reviewing: to provide precise information on how the analyses were done.

The nature of the SI file is 'different', because its goal is not to rewrite the manuscript but to provide the exact details with which interested readers can repeat and follow the steps of the results. To our defense of the 'challenge to present the method and results': the manuscript was critically read by two of our colleagues and by a professional English-writing scientist, with the help of whom we restructured the text and the methods to precisely follow every description of experiment, figure legend, corresponding method section and SI material. We believe this is the proper balance that is needed to present this complex material.

Action: We propose (if this is allowed by the journal standard) to add not just a reference to 'Supplementary methods' in the main text, but to include the appropriate section, to facilitate

looking up reference details. We have included this in the revised manuscript with the risk that the journal asks us to remove it again.

- The section "similarity assessment ... CellCognize classes" (line 320-377) remains unclear. What do the changes in probability distribution imply? What does the classification similarity mean? What do these values indicate? I tried to look this up in the Supplementary Methods, but still find the information insufficient.

Reply: The reviewer raises a rightful question here, to which we don't have the answer yet. We explain how the procedure works in the sense of producing a probability score of assignment to a class, and we further show that the difference of this probability score and the score with a true standard can be used to calculate the similarity. We demonstrate this further on an isolate from lake water that we sequence and compare in ANN.

What this similarity 'implies' we don't exactly know, because there is not yet sufficient experience with this type of calculation to link the ANN similarity score to e.g., a difference in genome sequence or some phenotypic difference. The goal of presenting it here was to show that it is not just about assigning to a class, but that the method allows a potential further differentiation based on similarity scores (and to provide the background calculation on how this can be done). This should be the basis for further advancing, right?

Action: We explain and discuss this in the Discussion lines 386-396, essentially giving the same arguments as above. As we conclude in line 394: we need to better understand how to use the similarity score by gaining more experience with new and other isolates and standards. We don't see how we can improve this further at this point.

- Please check the used machine learning terminology in the manuscript. For example:

→ An ANN is constructed by a set of linear equations, but is a single classifier.

Reply: We have checked again the terminology carefully. The outcome of the ANN is the classifier. See the Glossary of terms that we introduced.

Action: We removed the 'plural' term classifiers to the singular 'classifier' in l. 106/106 (Although we repeated the process thus having effectively multiple classifiers)

→ 'Predicted classification': either you predict or classify the label (cell type in this case) of unknown instances, but 'predicted classification' does not make sense.

Reply: With all respect, we use predicted classification, because it comes with a probability scoring that contrasts the 'true' classification with a 'predicted' one. See the 'Glossary of terms', where we define this and other terms according to good practice.

Action: we specified again in l. 106 and l. 123 that terms are explained in the Glossary and then used as such.

→ 'machine-learned ...' is not the way to express that a method belongs to the field of machine learning. For example, you perform 'supervised classification' or you use a 'machine learning algorithm'.

Changed in L. 28, l. 177 and 399

→ L81: None of these references (16,27,28,30) perform clustering. Reference 16 and 28 perform single-cell classification, in the same way as CellCognize. Reference 27 performs cytometric fingerprinting, and clusters the fingerprints at the community-level. Reference 30 calculates total cell counts, and proposes this as a methodology to go to absolute abundances by multiplying this number with taxon abundances.

We changed this line to 'FCM fingerprinting and classification methods'. We mentioned the single cell classification of those references in lines 67-72 already.

- I tried to access the Zenodo repository, but I was not able to.

Reply: As we explained in the accompanying letter, the link to Zenodo has not been activated yet, because it needs the journal name. The reviewer may have missed this. The link will be active upon acceptance (and in any case by June 30, 2020).

Other minor remarks:

Thanks for clarifying the intended use of cell type diversity. Maybe it's a good idea to also include this in the title?

Good idea: we changed to 'microbiota cell type diversity'

I am not sure, or maybe I am misunderstanding the experimental setup, whether the experiment described in lines 209-225 is the correct setup. Standards are added to an unknown background of cells, but, as you perform single-cell classification, it makes sense that these standards are recognized, as they are classified independently from other cells. In case I am misunderstanding, could you please improve the description of the experiment?

Reply: The reviewer is correctly interpreting this experiment. Single cells are classified, but then all the information is grouped to calculate the recovery per class. In presence of unknown background, it may be that other cells are included in such groups and falsely classified. This is what we wanted to show here.

Action: No further action needed.

Reviewer #2 (Remarks to the Author):

The authors answered the questions sufficiently enough. I have no further questions.